# Multi-species inversion and IAGOS airborne data for a better constraint of continental scale fluxes

Fabio Boschetti[1], Valerie Thouret[2], Greet Janssens Maenhout[3], Kai Uwe Totsche[4], Julia Marshall[1], Christoph Gerbig[1]

[1]Max Plank Institute for Biogeochemistry, Jena, 07745, Germany
[2]Laboratoire d'Aerologie, CNRS and Universite' Paul Sabatier, Toulouse, 31400, France
[3]European Commission Joint Research Centre, Institute for Environment and Sustainability, Ispra, 21027, Italy
[4]Friedrich Schiller University, Jena, 07743, Germany

*Correspondence to*: Fabio Boschetti (fabosk@bgc-jena.mpg.de)

**Abstract.** Airborne measurements of $CO_2$, CO, and $CH_4$ proposed in the context of IAGOS (In-service Aircraft for a Global Observing System) will provide profiles from take-off and landing of airliners in the vicinity of major metropolitan areas useful for constraining sources and sinks. A proposed improvement of the top-down method to constrain sources and sinks is the use of a multispecies inversion. Different species such as $CO_2$ and CO have partially overlapping emission patterns for given fuel-combustion related sectors, and thus share part of the uncertainties, both related to the a priori knowledge of emissions, and to model-data mismatch error. We use a regional modeling framework consisting of the Lagrangian particle dispersion model STILT (Stochastic Time-Inverted Lagrangian Transport), combined with high resolution (10 km x 10 km) EDGARv4.3 (Emission Database for Global Atmospheric Research) emission inventory, differentiated by emission sector and fuel type for $CO_2$, CO, and $CH_4$, and combined with the VPRM (Vegetation Photosynthesis and Respiration Model) for biospheric fluxes of $CO_2$. Applying the modeling framework to synthetic IAGOS profile observations, we evaluate the benefits of using correlations between different species' uncertainties on the performance of the atmospheric inversion. The available IAGOS CO observations are used to validate the modeling framework. Prior uncertainty values are conservatively assumed to be 20%, for $CO_2$ and 50% for CO and $CH_4$, while those for GEE (Gross Ecosystem Exchange) and respiration are derived from existing literature. Uncertainty reduction for different species is evaluated on a domain encircling 50% of the profile observations' surface influence over Europe. We found that our modeling framework reproduces the CO observations with an average correlation of 0.56, but simulates lower mixing ratios by a factor 2.8, reflecting a low bias in the emission inventory. Mean uncertainty reduction achieved for $CO_2$ fossil fuel emissions is roughly 38%; for photosynthesis and respiration flux it is 41% and 44%, respectively. For CO and $CH_4$ the uncertainty reduction is roughly 63% and 67%, respectively. Considering correlation between different species, posterior uncertainty can be reduced by up to 23%; such reduction depends on the assumed error structure of the prior and on the considered timeframe. The study suggests a significant uncertainty constraint on regional emissions using multi-species inversions of IAGOS in situ observations.

## 1. Introduction

As widely recognized at the international level, there is a need for reduction in anthropogenic emissions (IPCC). This however implies the necessity for reliable climate predictions from atmospheric models in order to allow policymakers to take informed decisions. Unfortunately, current climate predictions are hampered by excessive uncertainties; for example intercomparisons of different models show important differences on their predictions as shown in Friedlingstein (2016). This makes it difficult to assess the better environmental policies to implement. Because most biogenic fluxes in Europe are influenced by human activities, with 22% of Europe's land is dedicated to agriculture (FAO, 2013) and 45 % covered by forests, of which 80% are managed for wood supply (UNECE, FAO, 2011), understanding and managing these biogenic fluxes must also be a component of any policy to reduce anthropogenic emissions.

A commonly used approach to estimate carbon budgets by teasing apart sources and sinks in a given spatial domain is the atmospheric Bayesian inversion. Atmospheric inversions combine prior knowledge from emission inventories with atmospheric observations acting as a top-down constraint to produce better posterior knowledge. As the main goal of this study is to assess the benefit of inter-species correlations in reducing the uncertainty of the posterior state space, we are particularly interested in the effects of such correlations on the uncertainty reduction, defined as the difference between prior and posterior uncertainty normalized by the prior. The vast majority of published papers on atmospheric inversions investigate the budget of a single species, usually a long-lived greenhouse gas like $CO_2$ (e.g. Rödenbeck, 2003) or $CH_4$ (e.g. Hein, 1997; Bousquet, 2006), but the technique can also be applied to active species like CO (Bergamaschi, 2000). Note that carbon dioxide is a special case as atmospheric $CO_2$ mixing ratios result from a combination of strong anthropogenic sources with strong sources and sinks from biospheric processes, calling for a separation of anthropogenic from biospheric fluxes. One way to achieve such a separation is to measure CO alongside $CO_2$, and use CO as a proxy for $CO_2$ anthropogenic emissions. Several studies have made use the correlations among different species. One of the first example is the work from Enting (1995) on $CO_2$ and $^{13}CO_2$, while Brioude (2012) attempted to derive a $CO_2$ emission inventory without a prior emission estimate, instead using inventories of CO, $NO_y$ and $SO_2$. Similarly, Peischl (2013) made use of CO and $CO_2$ inventories to help quantifying sources of $CH_4$ in the Los Angeles basin. The ability of measuring multiple species has been proved useful also in remote sensing. For example, Pandey (2015) made use of simultaneously retrieved $CO_2$ and $CH_4$ total column to reduce scattering effect. Further examples of studies making use of co-emitted species can be found in the frame of atmospheric chemistry (Konovalov et al., 2014; Berezin et al., 2013; Pison et al., 2009). More focused on exploiting inter-species correlation to reduce uncertainty in the frame of Bayesian Inversion, Palmer (2006) made use of CO2-CO correlations to improve an inversion using data from the TRACE-P aircraft mission, while Wang (2009) employed a similar method using satellite data, obtaining a reduction in the flux error of a $CO_2$ inversion.

So far the lion's share of the studies investigating atmospheric inversions make use of both continuous in situ and flask measurements from ground based observational networks of tall towers (e.g. Kadygrov, 2015; Sasakawa, 2010). However, as profiles collected from an aircraft easily exceed the height of towers, airborne data may also prove an interesting option for this application. This alternative was tested in some recent studies that made use of aircraft profiles alone or in combination with other data sources (e.g.: Brioude, 2013; Gourdji, 2013). Methods to maximise the cost-effectiveness of airborne data are the use of unmanned aircraft (drones) and commercial airliners. The latter, in particular, allows for collecting data on a regular basis without requiring a particularly small or light sensor. The most important projects making use of commercial airliners are CONTRAIL (Comprehensive Observation Network for Trace Gases) (Machida, 2008), and MOZAIC/IAGOS (Measurements of Ozone and water vapor by in-service AIrbus aircraft / In-service Aircraft for a Global Observing System) (Marenco, 1998; Petzold, 2015). Both projects have been running for more than two decades and have produced extensive datasets that have proven to be important in the fields of atmospheric modeling and satellite calibration and validation (Zbinden et al.,2013; Sawa et al., 2012). Regarding carbonaceous species, CONTRAIL has so far been collecting $CO_2$ mixing ratio measurement, while IAGOS was focused on CO. In the next years the IAGOS fleet will simultaneously provide CO, $CO_2$ and $CH_4$ atmospheric concentration measurements (Filges, 2015), enabling the use of multi-species synergy in modeling applications. This synergy follows the fact that the collocated measurements share the same atmospheric transport and have partially correlated emission uncertainties.

This paper is focused on investigating the benefits on uncertainty reduction of such a multi-species inversion in comparison with a single-species inversion. To attain this goal, we set up a synthetic experiment utilizing the measurement times and locations collected from the IAGOS projects in the year 2011. The present paper is intended to pave the way for future studies making use of multi-species IAGOS datasets when they become available. A receptor-oriented framework was set up to derive flux interactions between the atmosphere and the biosphere using IAGOS data. The modeling framework is composed of a Lagrangian Particle Dispersion Model (LPDM, specifically the STILT model), a diagnostic biosphere-atmosphere exchange model (the VPRM model), gridded emission inventories, global tracer transport model output that provides the tracer boundary conditions for the regional domain, and a Bayesian inversion scheme. The present work is based on the modeling framework used in Boschetti (2015) and builds upon that by adding other species, and using a formal Bayesian inversion. A multi-species inversion was carried out in order to exploit the correlations in uncertainties between $CO_2$, CO, and $CH_4$, specifically in their respective uncertainties in a priori anthropogenic emissions and in model representation error. The aim of this multi-species inversion is to provide better estimates of anthropogenic emissions, and, in the case of $CO_2$, to better separate the biospheric from anthropogenic contributions. This paper is structured as follows: a short description of the different components of the modeling framework is given in Sect. 2; in Sect. 3 we present and discuss our results; Sect. 4 gives the conclusions.

## 2. Material and Methods

### 2.1 Modeling framework

Before describing the different models composing the modeling framework, we introduce some specific terminology to reduce ambiguity in Sect. 2.1.1-2.1.6. Quantities that can be observed are termed *species*, or *trace gases*, corresponding in this case to total $CO_2$, CO and $CH_4$. These three species are simulated using five *modeled species*, namely $CO_2$ from fossil fuels, $CO_2$ related to GEE (Gross Ecosystem Exchange) and to respiration, CO, $CH_4$. Modeled species related to anthropogenic emissions are modeled as the sum of contributions from different *emission sectors* (Table 1) and *fuel types* (Table 2); as a further factor of discrimination, both anthropogenic and biospheric contributions are split into monthly contributions. Simulated fluxes specific for different modeled species, emission sectors, fuel types and months of the year are called *flux categories*. In this Material and Methods section, a brief description of the different models that make up the modeling framework is given. For more detailed information, see Boschetti (2015).

### 2.1.1 Vertical profile input data

In this study the modeled profiles have the identical structure to those collected from the IAGOS fleet of commercial airliners. More precisely, the spatial and temporal coordinates of different observations will be used as input for the modeling framework whereas the observed values of atmospheric mixing ratios of CO and meteorological parameters themselves will play a role in calibrating the modeling framework.

Central for this work is the concept of the Mixed Layer (ML), the lower part of the troposphere in which trace gases are well mixed due to turbulent convection in the time scale of an hour or less, and in which the effect of regional surface-atmosphere fluxes is the strongest. As input to the inversion we use the enhancement of the species' mixing ratio within the mixed layer

relative to that in the free troposphere (FT), similar to the approach described in Boschetti (2015). This mixed layer enhancement best reflects the influence of regional fluxes. To compute this, we take the average mixing ratio within the mixed layer and subtract the value taken at 2 km above the mixed layer top ($z_i$), i.e. well within the free troposphere. The $z_i$ is a very important parameter in atmospheric modeling, and accounts for most of the transport uncertainty in the vertical

domain. In fact, assuming that the mixed layer is the part of the troposphere in which trace gases are well mixed due to turbulent convection, given a certain amount of trace gas in the ML, its mixing ratio depends strongly on its depth $z_i$. More precisely, even if the model has correctly reproduced the amount of trace gas in the real mixed layer, if the modeled $z_i$ is lower (higher) than the actual one, then the simulated ML mixing ratio will be higher (lower) than it actually should be. In the present study, modeled $z_i$ are corrected according to Boschetti (2015, Sect. 2.2.1)

### 2.1.2 Transport-flux coupling

The modeling framework is composed of a regional transport model (STILT), the EDGAR (Emission Database for Global
Atmospheric Research) emission inventory to model anthropogenic emissions, VPRM (Vegetation Photosynthesis and Respiration Model) to model emissions from the biosphere and output from global transport models for lateral boundary conditions for the different modeled species. The expressions 'anthropogenic emissions' and 'fossil fuel emissions' are considered synonymous in this paper and are used to indicate the sum of fossil fuel and biofuel emissions, without including contributions from LULUCF (Land Use, Land use Change and Forestry).

For regional transport we make use of the LPDM STILT (Stochastic Time-Inverted Lagrangian Transport) (Lin, 2003) to derive the sensitivity of the atmospheric mixing ratio measurement to upstream surface-atmosphere fluxes, so-called "footprints". Briefly, for each measurement location and time (also called receptor point), the model releases an ensemble of virtual particles that are driven back in time using wind fields from ECMWF and turbulence as stochastic process; the
residence time within the lower half of the mixed layer is used to determine the potential contribution from surface fluxes, and the cumulative sum of these contributions determines the footprint, that identifies the part of the domain with a certain influence on a single receptor point. To represent the mixed layer enhancements, the footprints for receptors within the boundary layer are averaged, and the footprint for the free tropospheric receptor is subtracted from this, resulting in a footprint for the mixed layer enhancements. This footprint is then matrix-multiplied with an emission map from an emission
inventory, resulting in a simulated mixing ratio enhancement corresponding to the regional contribution at the measurement location.

A detailed description of STILT is given in Lin (2003) and Gerbig (2003). We use STILT coupled with emission models for both anthropogenic (EDGAR) and biosphere (VPRM) fluxes on a regional domain that covers most of Europe (33° to 72° N,

-15° to 35° E) with a spatial resolution of 1/8 degree for latitude and 1/12 degree for longitude, roughly corresponding to 10 km. As lateral boundary condition for CO mixing ratios the MACC reanalysis (Inness, 2013, downloaded from http://www.ecmwf.int) was used, whereas for $CO_2$ and $CH_4$ we use output from the Jena CarboScope (Rödenbeck, 2003; $CO_2$ data available from www.bgc-jena.mpg.de/CarboScope/) which is based on forward simulations of global-inversion optimized fluxes with the TM3 transport model (Heimann and Körner, 2003). TM3 fields have lower resolution, but they are chosen for their consistency with measurements from the ground-based network. In addition, spatial resolution is of relatively minor importance for the contribution from the lateral boundary as it is far away from the measurement locations.

For fossil fuel emissions we use a model based on the EDGAR emission inventory modified following the same approach taken for COFFEE ($CO_2$ release and Oxygen uptake from Fossil Fuel Emission Estimate) (Steinbach, 2011,; Vardag, 2015). More precisely, to obtain hourly resolved emissions from the original EDGAR annual fluxes for different emission categories we add specific temporal activity factors (Denier van der Gon, 2011) to account for differences in emissions due to seasonal, weekly and daily cycles. In addition, the different emission categories are further split into contributions from different fuel types from British Petroleum's Statistical Review of World Energy 2014 (BP, 2014). The World Energy Outlook from IEA as alternative source of information was not chosen, as the report from BP is available earlier (April vs. November of the following year). This allows for taking into account changes in emissions between different years. Such an emission model provides hourly resolved fluxes for each fossil fuel flux category with a spatial resolution of roughly 10 km on our regional European domain. For each of the three anthropogenic modeled species ($CO_2$, CO and $CH_4$), different emission maps are used as input. Temporal profiles are then applied to these sector- and fuel-specific emission maps. To take into account also the contribution from the biosphere we use the Vegetation Photosynthesis and Respiration Model (VPRM). VPRM simulates realistic patterns at small spatial (10 km x 10 km) and temporal (hourly) scales and is used here to provide the a priori fluxes for biosphere-atmosphere exchange of $CO_2$. This model is described in detail in Mahadevan (2008).

STILT transport is driven by meteorological fields from the ECMWF IFS (12 hour forecasts twice daily at 3-hourly temporal resolution), which have a spatial resolution of 0.25 degree with 61 vertical levels. In the following, we will refer to the STILT/EDGAR/VPRM/MACC/TM3 combination of transport, simulated fluxes and advected boundary conditions as merely 'STILT' for simplicity.

**2.1.3 Bayesian inversion**

Atmospheric inversions provide an estimate of the distribution of sources and sinks over the domain's surface from available concentration measurements ("top-down" approach). This can be formalized in the following linear relation:

$$\boldsymbol{y} = \mathbf{K}\boldsymbol{\lambda} + \boldsymbol{\varepsilon} \tag{1}$$

Where the $y$ vector contains the $n$ observations, and $\mathbf{K}$ is the Jacobian matrix that relates the observations with the state vector $\lambda$. In the present study the focus will be on surface-atmosphere gas exchanges due to both biospheric processes and anthropogenic emissions. So the observations are trace gas mixing ratios at different times and locations, $\mathbf{K}$ is the product of a transport operator $\mathbf{H}$ that maps flux sensitivities at different times and locations with a set of gridded fluxes $\mathbf{F}$ for the categories of interest, while the state vector $\lambda$ contains the $m$ scaling factors for the flux categories of interest. $\mathbf{H}$ has $n$ rows and a number of column equal to $h=N_x{*}N_y{*}N_t{*}N_s$ being respectively the number of pixels in the emission model along the x and y axes, the number of (hourly) simulations in the whole year of interest and the number of state vector elements, resulting in a huge matrix. As the matrix $\mathbf{F}$ describes the different simulated gridded fluxes, it is comparably large and has $h$ rows and $m$ columns. By considering K as the result of the product of these two large matrices, it is possible to limit its dimensions to only $n$ rows and $m$ columns; this allows for simplifying the critical task of relating observation with simulated fluxes of the categories of interest. The state vector accounts for specific emission sectors (Table 1) and fuel types (Table 2) for each one of the three modeled species from the EDGAR emission model, plus gross fluxes (gross ecosystem exchange GEE and respiration R) modeled by VPRM for 5 different vegetation classes. For both anthropogenic and biospheric fluxes the temporal resolution of the state vector is monthly. The number of state vector elements per month amounts to 69 scaling factors for the different fuel- and sector-specific anthropogenic emissions for each species, and 10 scaling factors for biosphere-atmosphere exchange (respiration and photosynthesis for each of the five vegetation classes), so in total 217 scaling factors per month, or 2604 scaling factors for the full year. To avoid large memory requirements for $\mathbf{H}$ and $\mathbf{F}$ matrices, their product is directly computed within the STILT code. The random error $\varepsilon$ accounts for measurement error related to uncertainty in the observation and to model-data mismatch resulting from model uncertainty.

Bayesian inversion combines observations (IAGOS profiles) with a priori information (scaling factors and their a priori uncertainties) to reconstruct the most probable state vector. Optimum posterior estimates of the scaling factors are obtained by minimizing the following cost function $J$ (Rodgers, 2000):

$$J(\lambda) = (y - \mathbf{K}\lambda)^T \mathbf{S}_\varepsilon^{-1}(y - \mathbf{K}\lambda) + \left(\lambda - \lambda_{prior}\right)^T \mathbf{S}_{prior}^{-1}(\lambda - \lambda_{prior}) \tag{2}$$

Here the first and the second term are the observational constraint and the prior constraint term respectively. The prior scaling factors for the fluxes of the different tracers are set equal to one. $\mathbf{S}_\varepsilon$ is the error covariance matrix for the mismatch between simulated and observed mole fractions (model-data mismatch) and accounts for instrumental uncertainty, uncertainty related to the transport model, and other sources of uncertainty like boundary conditions and flux aggregation not accounted for through the state vector adjustment. $\mathbf{S}_{prior}$ is the error covariance matrix for the prior scaling factor; its implementation requires a different approach for biospheric and anthropogenic fluxes. The detailed error structure for model-

data mismatch and prior uncertainty is described in the Sect. 2.1.4. Minimizing the cost function results in an optimal posterior estimate of the state vector $\lambda$ that is consistent with both the measurements and the prior model estimates:

$$\hat{\lambda} = \left(\mathbf{K}^\mathbf{T}\mathbf{S}_\varepsilon^{-1}\mathbf{K} + \mathbf{S}_{\mathbf{prior}}^{-1}\right)^{-1}\left(\mathbf{K}^\mathbf{T}\mathbf{S}_\varepsilon^{-1}y + \mathbf{S}_{\mathbf{prior}}^{-1}\lambda\right) \tag{3}$$

The error covariance matrix of the optimal posterior state (the posterior uncertainty) is given by:

$$\mathbf{S}_{\mathbf{post}} = \left(\mathbf{K}^\mathbf{T}\mathbf{S}_\varepsilon^{-1}\mathbf{K} + \mathbf{S}_{\mathbf{prior}}^{-1}\right)^{-1} \tag{4}$$

Note that this quantity depends on neither the prior fluxes nor the measured mixing ratios, but only on their respective uncertainties and on the transport matrix $\mathbf{K}$. In this study, the inverse of the matrices was calculated using the R-function 'solve' from the base package of R version 3.0.0 (http://www.r-project.org/).

The targeted quantities of this study are the aggregated emissions over a specific area at a specific time scale (e.g. month); those quantities can be derived from the prior and posterior state through a spatiotemporal aggregation operator $\mathbf{A}$ that allows for the conversion of scaling factors into physically representative quantities. As the pseudo-observations are clustered around a single location (Frankfurt), fluxes over the whole European domain can very likely not be constrained. Therefore, as a spatial aggregation scale we chose an area from which fluxes have a significant contribution to the observations made at Frankfurt. For this we compute the temporally accumulated footprint values (cf. Sect. 2.1.2) for the whole year 2011, and select those spatial pixels that correspond to 50% of the total (spatially integrated) footprint (Fig. 1). Note that by using this aggregation scale we assume perfectly-known distribution within a given flux category that can result in aggregation error, especially with respect to biogenic fluxes, that are not so well known as anthropogenic fluxes. However, the chosen domain of aggregation is quite small, and the total anthropogenic fluxes are divided according to species, emission categories, fuel types and months. This result in 69 degrees of freedom per month for each anthropogenic species and 10 degrees of freedom per month for the biospheric fluxes; for this reason we expect the aggregation error not to be a particularly important source of uncertainty. The prior and posterior uncertainty of these targeted quantities ($\sigma_{prior}$ and $\sigma_{post}$) is obtained by applying the aggregation operator to the respective uncertainty covariances:

$$\sigma_{prior} = \sqrt{\mathbf{A}^\mathbf{T}\mathbf{S}_{prior}\mathbf{A}} \quad \text{and} \quad \sigma_{post} = \sqrt{\mathbf{A}^\mathbf{T}\mathbf{S}_{post}\mathbf{A}} \tag{5}$$

Different versions of the aggregation operator were created for this: emissions categories are aggregated according to different fuel types (coal, oil, gas, bio, waste, other) and according to emission sectors (energy, transport, industry, buildings,

agriculture, waste, fossil fuel fires). Note that only these aggregated fluxes are optimized, not the individual gridded fluxes of the emission inventories.

To quantitatively assess the information provided by the inversion, the reduction of uncertainty in the posterior compared to
the prior estimate is a useful measure. The uncertainty reduction *UR* is defined as:

$$UR = 1 - \frac{\sigma_{post}}{\sigma_{prior}} \qquad (6)$$

The uncertainty reduction ranges from 0 (posterior as large as the prior uncertainty) to 1 (posterior negligible compared to
the prior uncertainty).

### 2.1.4 Prior error structure

As in this study a multi-species inversion with CO, $CO_2$ and $CH_4$ is envisioned, we have the chance to exploit the correlations in the uncertainties of the different trace gases related to both a priori fluxes and model-data mismatch. This is
particularly true for CO and $CO_2$ because they share a larger part of the emission sources, which implies correlations in the respective uncertainties. In the multi-species inversion, such information is stored in the areas of the error covariance matrices that describe covariance between different modeled species (off-diagonal 'blocks' in Fig. 2b for $S_{prior}$ and Fig. 3b for $S_\varepsilon$). In the single-species inversions, said covariance is set to zero, corresponding to a situation where the different species are completely independent of one another. Conversely, the measurement uncertainty is stored in the main diagonal
of the $S_\varepsilon$ (Fig. 3d).

We used a single year (2011) dataset restricted to the vertical profiles centered at the Frankfurt airport, and restricted to daytime during well-mixed atmospheric conditions (10:30 to 17:30 CET). The dataset contains 1098 pseudo-observations, 366 for each of the three observable species, whereas the state vector contains the scaling factors for 2604 flux categories,
each equal to one in the prior.

The prior error covariance matrix can be expressed as follows:

$$\mathbf{S_{prior}} = \mathbf{C_{prior}}\boldsymbol{\rho_{prior}} \qquad (7)$$

where $\mathbf{C_{prior}}$ is the prior error correlation matrix (Fig. 2a) and $\boldsymbol{\rho_{prior}}$ is a prior rescaling matrix described in Sect. 2.1.5 (Fig. 4a). First we describe how $\mathbf{C_{prior}}$ is generated. The prior error correlation matrix is a square matrix of rank 2604, reflecting

the length of the state vector, and results from the product of three components (Fig. 2b, 2c and 2d) accounting for correlations between flux categories according to the modeled species, emission sectors and fuel types respectively. In four different instances, a correlation of 0.7 is applied:

1. Between different anthropogenic modeled species
2. Between GEE and respiration
3. Between different emission sectors
4. Between different fuel types

Such a correlation implies that the explained variance for each constraint everything else being equal is roughly 50%, (0.7 to the square equals 0.49) with the rest remaining independent. In addition, the correlation between fossil-fuel-related and biosphere-related scaling factors is zero, and the same holds for fluxes of different months, indicating complete independence from one another. In this study, we assume a certain annual total domain wide flux uncertainty, and then break it down by sectors, fuels, and months by inflating the error. By assuming no correlation between different months we ensure maximum flexibility in the system to retrieve month-to-month changes based on the observations. Assuming correlation between months would be possible, but has not been investigated here. It is unclear how good the seasonal variation in emissions from the inventories actually is, so in order to not rely too much on these we chose zero correlation. Investigating the effects of different correlation set-ups for the seasonal cycle could be the focus of future research.

### 2.1.5 Prior error scaling

After having specified the prior error correlation matrix $\mathbf{C_{prior}}$, we now describe how we rescale it to obtain $\mathbf{S_{prior}}$; for this task we rewrite Eq. (7) as

$$\mathbf{S_{prior}} = \mathbf{C_{prior}}\boldsymbol{\rho}_{\mathbf{prior}} =$$

$$(8)$$

$$= \begin{pmatrix} C_{11} & C_{12} & C_{13} & 0 \\ C_{21} & C_{22} & C_{23} & 0 \\ C_{31} & C_{32} & C_{33} & 0 \\ 0 & 0 & 0 & C_{bio} \end{pmatrix} \begin{pmatrix} 1/\rho_1\rho_1 & 1/\rho_1\rho_2 & 1/\rho_1\rho_3 & 0 \\ 1/\rho_2\rho_1 & 1/\rho_2\rho_2 & 1/\rho_2\rho_3 & 0 \\ 1/\rho_3\rho_1 & 1/\rho_3\rho_2 & 1/\rho_3\rho_3 & 0 \\ 0 & 0 & 0 & \rho_{bio}^2 \end{pmatrix}$$

where each $\mathbf{C_{ij}}$ is a subset of the fossil fuel part of $\mathbf{C_{prior}}$ ('block') as shown in Fig. 2, and each $\rho_i$ is defined as

$$\rho_i = \sqrt{\frac{A_i'^T C_{ii} A_i'}{\left(\sum_j A_{ij}' \, \varepsilon_i\right)^2}} \tag{9}$$

where **A'** is the aggregation operator for annual fluxes over the full domain, and $\varepsilon_i$ is the corresponding relative prior uncertainty, assuming the values specified in Table 3 for different cases. Case 1 is considered as the default case, with prior uncertainty values conservatively assumed to be 20%, for $CO_2$ and 50% for CO and $CH_4$. Conversely, **C$_{bio}$** covers the biosphere part of **C$_{prior}$**, and for $\sum A_i' \varepsilon_i$ for $\rho_{bio}$ we use a prior uncertainty of 0.54 GtC y$^{-1}$, as derived in Panagiotis (2016) for the VPRM model. The biospheric part of the prior error covariance matrix assumes no correlation with the fossil fuel species.

The posterior of each Bayesian inversion depends on its specific prior. As the multi- and single-species inversion have different prior uncertainty structures, the uncertainty reduction for targeted quantities cannot be directly compared (Eq. (4)). To be able to compare the two inversions, we require that the a priori aggregated uncertainty of the targeted quantities remains the same, and distribute it differently each time; the prior rescaling matrix **ρ$_{prior}$** is needed for this task. The benefits were tested for observations taken in different months and for three different error structures in the prior uncertainty. As a priori aggregated uncertainty we use a percentage of the aggregated modeled emissions for fossil fuels across the whole year. Table 3 shows the percentage values used for different cases.

### 2.1.6 Model-data mismatch error structure

In an atmospheric inversion, the model-data mismatch from every uncertainty source (such as measurement uncertainty, transport model uncertainty, spatial representation error due to limited model resolution, and boundary condition inaccuracies) needs to be taken into account. In our inversion scheme, we parameterize both the transport model uncertainty and the measurement uncertainty, with the latter playing a minor role. The model-data mismatch covariance matrix (**S$_\varepsilon$**) is constructed according to the following equation:

$$\mathbf{S_\varepsilon} = \mathbf{C_s C_t} \varepsilon_{tran}^2 + \varepsilon_{meas}^2 \tag{10}$$

where **C$_s$** accounts for correlations between different observed species (Fig. 3b), **C$_t$** accounts for the temporal correlation (Fig. 3c), $\varepsilon_{tran}$ is the total transport error and $\boldsymbol{\varepsilon}_{meas}^2$ accounts for all of the non-transport related errors like spatial representation error and lateral boundary conditions (Fig. 3d).

The assumed measurement uncertainty is 1 ppm for $CO_2$, 20 ppb for CO and 20 ppb for $CH_4$, while $\varepsilon_{tran}$ is time dependent and assumed to be proportional to the modeled enhancement due to regional fluxes. The assumed measurement uncertainty is higher than the expected instrument precision because it also includes in addition the uncertainties related to spatial representation and lateral boundary condition. $\varepsilon_{tran}$ is characterized as follows by different components in the vertical and horizontal domain:

$$\varepsilon_{tran} = enh \sqrt{\left( \varepsilon_{tran\_h}^2 + \varepsilon_{tran\_v}^2 \right)} \tag{11}$$

where *enh* indicates the modelled enhancement, and both the horizontal transport error $\varepsilon_{tran\_h}$ and the vertical transport error $\varepsilon_{tran\_v}$ are characterized as percentage error; $\varepsilon_{tran\_h}$ is assumed to be 50%, while $\varepsilon_{tran\_v}$ is a profile-specific relative error with a mean value of about 10%. The vertical transport error accounts for the fact that the shallower the mixed layer is, the more difficult it is to model the atmosphere. We assume that after zi-correction the remaining error is on the order of 50 m (related to the vertical resolution of the profile data), so the relative error $\varepsilon_{tran\_v}$ is assumed as the ratio of 50 m to the modeled $z_i$; in this way we obtain an error that gets larger the shallower the mixed layer is. For the horizontal component, an uncertainty of 50% is a conservative estimate based on Lin and Gerbig (2005), where the horizontal transport error is found to be 5.9 ppm for $CO_2$. This, combined with about 10 ppm of drawdown in the mixed layer relative to the free troposphere, gives something like 50% error in the regional flux signal. The vertical component is so much smaller in percentage since the simulated mixing ratios are already corrected for mismatch between modeled and observed $z_i$.

In the multi-species inversion, the transport error correlation across species is 0.7 (Fig. 3b), while in the single-species inversion this is set to zero. Time correlation is assumed to decay exponentially with an exponential constant of 12 hours. The between-species correlation for model-data mismatch related to transport uncertainty reflects the fact that species are partially co-emitted and share the same atmospheric transport (and its related uncertainty).

## 2.2 Synthetic experiment

### 2.2.1 Pseudo-data generation

As explained in the introduction, in situ measurements are not available for all of the three trace gases of interest, but only for CO. For this reason this paper aims to evaluate the benefits of a multi-species inversion over a corresponding single-species one by performing a synthetic experiment, using pseudo-observations derived by perturbation of the model outputs based on a priori state vector values. More precisely, the pseudo-observation vector is obtained by matrix multiplication between the Jacobian matrix **K** and what we assume to be the true state vector. The true state vector itself is obtained by using the sum of the prior state vector (all values equal to one) and a random realization of the prior error, truncated to avoid

negative state vector values. In detail, the error realization is obtained by multiplying a randomly generated, normally distributed vector with the prior error covariance matrix. This ensures that such realization has the same error correlation of the prior uncertainty. Where the result of such matrix-vector product is negative, the same operation is performed recursively until all elements of the state vector are positive. This ensures that the difference between the true and prior state vector has the same error correlation structure as described by the prior error covariance matrix.

## 3. Results and Discussion

Before evaluating the performance of the inversion scheme in reducing the uncertainty of the state space, a closer look at the ability of the modeling framework to reproduce the enhancements is necessary. Unfortunately, this can be done only for CO as actual measurements are not available for the other species. Figure 5 shows the mean daily enhancement of the three fossil fuel species for both observations and model outputs using prior emissions. A common feature to the three trace gases is that lower values tend to occur during summer time due to a better mixing of the atmosphere. Conversely, enhancement values tend to be higher during winter, reflecting the more stratified atmosphere of the cold months.

In Fig. 5 the modeled CO plot was multiplied by a factor of 2.8, corresponding to the mean ratio between observed and modeled CO enhancements, similar to what was found in Boschetti (2015). Mixing ratio values are highly variable, but the model produces a good indication of the temporal variation of the ML enhancement; the squared correlation coefficient between observed and modeled CO enhancements is 0.62, while the standard deviation of corrected model and observation residuals is 85 ppb; note that by not accounting for the $z_i$ correction, such values would be 0.56 and 87 ppb respectively. The median of the mixing ratio enhancement for the three trace gases is 2.8 ppm for $CO_2$, 18.6 ppb for CO and 26.6 ppb for $CH_4$. For $CO_2$ this seasonal difference is enhanced due to the simultaneous presence of both anthropogenic and biogenic emissions. During summer values are slightly negative due to strong photosynthesis fluxes from growing vegetation from the active combined with deeper vertical mixing. Negative values arise in 31% of the cases predominantly during the warmer months, implying that during the growing period uptake by photosynthesis dominates over release from combustion and respiration. Both CO and $CH_4$ experience higher values during winter due to the shallow mixed layer usually associated with cold temperatures, and lower values during summer as higher temperature cause the mixed layer to reach higher altitudes; differences related to seasonal domestic heating and transportation may also play a role. In addition, enhancement for both species is occasionally negative, most likely owing to advection of polluted air masses in the free troposphere. An alternative explanation is that strong winds at lower heights can disperse the emissions in the boundary layer and create a situation in which the mixing ratio in the FT is higher than in the ML.

Figure 6 shows the prior and posterior error covariance matrices for the base multi-species inversion. Note that $CO_2$ from anthropogenic emissions is assumed to be independent from biogenic emissions; therefore prior error correlation between these categories is zero. The posterior error covariance matrix for the multi-species inversion (Fig. 6b) shows lower values corresponding to an average uncertainty reduction of 23% across all state vector elements, while the posterior error covariance matrix for the single-species inversion (not shown) is characterized by a mean uncertainty reduction of 20%. This result implies that the multi-species inversion improves the uncertainty reduction by roughly 15%. Negative values in the posterior error correlation matrix are to be expected because different categories are bind together by correlations and therefore are not free to vary independently.

Figure 7 and 8 show a priori, a posteriori, and "true" fluxes related to different aggregated fuel types and to different emission categories as described in Tables 1 and 2 for the months of July and December. Figure 8 also shows the biospheric contribution (as absolute values) scaled down by a factor of 10. As is to be expected, the biospheric contributions show strong differences according to the seasonal cycle, while anthropogenic emissions remain rather stable. However, it is worth pointing out that while the fossil fuel prior is similar for both months, the assumed truth can be rather different due the random assignment of the prior error realization. In most cases, the posterior adapts and is therefore closer to the truth than the prior; the posterior uncertainty is also visibly reduced, as expected. Regarding the different tracers, $CO_2$ and CO show a somewhat similar pattern indicating a partial overlap in dominating emission categories while $CH_4$ is dominated by different contributions in both fuel types and emission categories.

Our modeling framework is currently not well suited to account for unreported sources of $CH_4$ due to the lack of information about natural gas and oil production operations, or from recent and old mining areas.. Many recent studies have discussed the problem, mainly referring to shale basins exploited via hydraulic fracturing in the US (e.g. Kort et al., 2016; Karion et. al, 2015; Lyon et al., 2015). For example, Karion (2015) concludes that EDGAR underestimates methane emissions associated with oil and gas industry by a factor of 5 in the US. However, the situation over the European continent may be quite different. In a review about risk assessment of shale gas development in the UK, Prpich (2015) reports that the European Union is generally much more cautious about unconventional oil and gas sources, while a recent study on a methane plume over the North Sea (Cain et al., 2017) concluded that the bulk signature of said plume originated from on-shore coal mines and power stations in the Yorkshire area.

In general, the absence of some emission sources in an inventory is equivalent to the assumption of having point sources not included in the emission map, but still contributing to the measurements. The inversion scheme would typically react to this by assigning such point sources in some other sector other fuel type. As a result, the posterior enhancements would be biased low in proximity of that point sources, and (slightly) biased high for influences from other regions with the same sector or

fuel type. This issue should definitely be considered in a future study making use of actual CO, $CO_2$ and $CH_4$ observations from IAGOS but has limited effects on this paper, as our main focus is on the benefits of inter-species correlation on the posterior uncertainty in the frame of a synthetic experiment

Note that our modeling framework does not allow for simulating CO biogenic fluxes during the growing season. Warm days in summer correspond to large amount of biogenic VOC's being emitted from the vegetation, producing CO to non-negligible levels. According to Hudman (2008), anthropogenic emissions accounts for only 31% of CO emissions in the US during summer. Conversely, according to estimates from EDGAR, CO anthropogenic emissions during summer are about 18% of the annual anthropogenic emissions. Combining these two results, one could conclude that CO production from

biogenic sources accounts for roughly 42% of total annual CO emissions.

In general, the absence of some emission sources in an inventory is equivalent to the assumption of having point sources not included in the emission map, but still contributing to the measurements. The inversion scheme would typically react to this by assigning such point sources in some other sector other fuel type. As a result, the posterior enhancements would be biased low in proximity of that point sources, and (slightly) biased high for influences from other regions with the same sector or

fuel type. This issue should definitely be considered in a future study making use of actual CO, $CO_2$ and $CH_4$ observations from IAGOS but has limited effects on this paper, as our main focus is on the benefits of inter-species correlation on the posterior uncertainty in the frame of a synthetic experiment.

$CO_2$ and CO are dominated by combustion sectors (Fig. 8). The most important emission sectors for $CO_2$ are energy,

industry, transport and building, each contributing 7-10 MtC month$^{-1}$ in July and 6-14 MtC month$^{-1}$ in December. Dominant fuels (Fig. 7) for $CO_2$ are coal, gas and oil, whose prior fluxes (pseudo data) have a magnitude of 6-11 Megatons of carbon per year (MtC month$^{-1}$) in July and 8-14 MtC month$^{-1}$ in December. For CO the most important emission sector is heating of buildings during winter contributing a 0.19 MtC month$^{-1}$ flux with only secondary contributions from industry and transport with a magnitude of 0.04 MtC month$^{-1}$ and 0.05 MtC month$^{-1}$ respectively (during July and December). The dominant fuel

for CO is biofuel with 0.19 MtC month$^{-1}$ emissions during winter. The secondary industrial and transport contributions originate in summer from oil and biofuels with a magnitude of 0.06-0.08 MtC month$^{-1}$ and from agricultural waste burning with a magnitude of 0.06-0.11 MtC month$^{-1}$.

Contrary to $CO_2$ and CO, $CH_4$ is determined by non-combustion sectors, more specifically by a contribution of 0.15 MtC

month$^{-1}$ flux from agriculture (manure management and rice cultivation) in July with secondary contributions from waste and energy with a magnitude of roughly 0.06-0.08 MtC month$^{-1}$ in both July and December. Other non-combustion sectors, in particular wastewater treatment and landfills contribute to a total of 0.16-0.24 MtC month$^{-1}$ of emissions. These non-combustion sectors contribute to less than 20% of total $CO_2$ emissions, with 1.13 MtC month$^{-1}$ from the cement and lime industry and less than 20% to the total CO emissions (0.03 MtC month$^{-1}$ from the metal industry).

The contribution to $CO_2$ from biospheric primary production (a sink for atmospheric $CO_2$) is about 100 MtC month$^{-1}$ in July, which drops to almost zero in December, while respiration values are 50 MtC month$^{-1}$ in July and roughly 15 MtC month$^{-1}$ in December.

As further assessment of the inversion performance, we tested the ability of the inversion scheme to capture the truth compared with a perturbed version of the prior. Such perturbed version is obtained by adding a random distribution with mean and standard deviation equal one to the prior state space, similar to how the truth is obtained. For each simulated species we calculated the total annual fluxes for prior, posterior, truth, and perturbed prior. From these total fluxes we then derive the overall residual between prior and truth, posterior and truth, and perturbed prior and truth. It is clear from Table 4
that while the overall bias between posterior and truth is lower than the prior-truth bias, the bias between perturbed prior and truth is much higher, implying that the performance of the inversion is not an artifact of the pseudo-data generation. In addition, it was found that the truth-posterior bias of the multi-species inversion is mostly slightly lower compared to the single-species inversion. Such difference is between -2.2% and 7.6%, according to the simulated species, with an overall value of 0.3%.

Improper characterization of the error correlation may result in systematic bias in the posterior estimate. As mentioned in Sect. 2.1.6, inter-species correlation, the correlation between different fuel types and the correlation between different emission sectors in $S_{prior}$ is assumed equal to 0.7 (Sect. 2.1.4). To assess how well the system will reproduce the 'true' fluxes with incorrectly specified correlations, a series of experiments was performed in which the inter-species correlation in $S_\varepsilon$
remains equal to 0.7, while the three correlation coefficients in $S_{prior}$ assume different values ranging from 0.1 to 0.9. Table 5 shows the residuals between total annual posterior fluxes and total annual true fluxes for the five simulated species, derived similarly as for Table 4. We found that for all species the uncertainty reduction increases with correlation. More precisely, from correlation 0.1 to 0.9, the annual uncertainty reduction for anthropogenic $CO_2$ increases from 26.6% to 51.7%, while the increase is lower for GEE (from 72.4% to 73.1%) and respiration (from 39.3% to 41.3%) because the biospheric fluxes
are independent from other species. For CO, the uncertainty reduction increases from 60.7% (with correlation 0.1) to 66.4% (with correlation 0.9). The annual uncertainty reduction for $CH_4$ increases from 60.5% to 67.5%.

In addition, the posterior-truth biases are always lower than the prior-truth biases. The posterior uncertainty values (1-sigma) are usually larger then the corresponding bias values as expected, except for CO and for $CH_4$ with prior correlations equal to
0.9. Thus the posterior is not significantly different from the truth. Conversely, the prior (not shown) is significantly different from the prior in the majority of cases for fossil fuel fluxes, and in some cases also for biogenic fluxes. The effect of assuming the incorrect error correlations appears to be in general small, possibly implying a relative robustness of our methods. Following this result, the fact that $CH_4$ is only partially co-emitted with $CO_2$ and CO should not affect the inversion in a strong way. For all of the experiments, the residuals between true and posterior fluxes are lower than residuals between

true and prior fluxes for each of the simulated species; the difference between the cases with maximum and minimum residuals is around 4.2%. In addition, we found that the posterior aggregated fluxes in the nine experiments are not significantly different from each other, implying that the system is fairly robust against errors in the assumed inter-species correlation.

Before investigating the benefits of correlations between different tracers, it is meaningful to evaluate the uncertainty reduction in the monthly budgets for all five modeled species (Fig. 9, based on targeted spatial domain in Fig. 1). The first thing to note is that for all of the five trace gases the posterior uncertainty is lower than the prior one, as it should be. In addition, prior uncertainty varies through the year, reflecting modulation in emission fluxes obtained by adding activity

factors to describe the seasonal, weekly and daily cycle.

Prior uncertainty assumes values around 0.4-0.6 MtC month$^{-1}$ for $CO_2$, 5-15 ktC month$^{-1}$ for CO, and 15 ktC month$^{-1}$ for $CH_4$. For GEE the prior uncertainty is between 0.3 MtC month$^{-1}$ and 46.7 MtC month$^{-1}$, and for respiration it is 5.1-19.0 MtC month$^{-1}$. Posterior uncertainty for $CO_2$ is 0.24-0.38 MtC month$^{-1}$ for fossil fuel emissions, 0.3-9.9 MtC month$^{-1}$ for GEE and 3.1-10.4 MtC month$^{-1}$ for respiration, while it has a range of 3.3-4.7 ktC month$^{-1}$ for CO and 2.7-7.0 ktC month$^{-1}$ for $CH_4$.

Mean uncertainty reduction of the monthly values is 38% for fossil fuels emission of $CO_2$, 41% for GEE, and roughly 45% for respiration, 63% for CO and about 67% for $CH_4$. It is worth pointing out that such values are higher than the mean uncertainty reduction in the scaling factors (23%); this happens because the most representative emission sectors are those influencing the observations the most and thus are also the most constrained.

In addition, note that in this case, the posterior uncertainties for single- and multi-species inversions are similar for the modeled species, with the exception of the $CO_2$ anthropogenic contributions. To generalize this last result, we tested the benefit of a multi-species inversion for the different cases of prior uncertainty values shown in Table 3. As an indicator for the benefit of including correlation between different species, we use the ratio between posterior uncertainty of the multi-species inversion and the posterior uncertainty of the corresponding single-species inversion. A value of one means that there

is not benefit in adding an inter-species correlation to the inversion, while values greater than one means that a multi-species inversion is even less constrained than a single-species one. We expect this indicator to be less than one, meaning that inter-species correlations actually improve the constraint power of the inversion. As before, we consider here the uncertainties of the retrieved budgets for the 50% footprint, where the surface influence is strongest (Fig. 1). Values of this uncertainty ratio for the different trace gases as function of month are shown in Fig. 10 for the different cases listed in Table 3. The benefit of

including inter-species correlations shown in Fig. 10 does not depend on different manifestations of the true fluxes, but only on the posterior uncertainty of the multi- and single-species inversions.

All of the species experience a reduction in the posterior uncertainty ratio due to the addition of inter-species correlation; said reduction is up to 20% for fossil fuel $CO_2$ and up to 10% for the other species; In addition, anthropogenic $CO_2$ is more

sensitive to the prior relative error values than CO and $CH_4$. As the uncertainty of GEE and respiration is not modified, they show little to no variations for different cases (Fig. 10). There is a dependence of the benefit of the multi- over a single-species inversion on the prior uncertainty values (differences between cases 1-3), with the largest difference for fossil fuel emissions of $CO_2$. Interestingly for case 2 with reduced prior uncertainty for fossil fuel $CO_2$ emissions the benefit nearly

doubles over the default case (Case 1). Also reducing the prior uncertainties of CO and $CH_4$ emissions (Case 3) more or less compensates for this increase in benefit. The reason for both of these results is probably to be searched in Eq. 8. In fact, changing the prior uncertainty in $CO_2$ emissions means to also change the off-diagonal blocks linking the different species together. However, by reducing the anthropogenic $CO_2$ uncertainty from 20% to 10% (Case 2), the diagonal block for $CO_2$ in the prior uncertainty changes by a factor four, while the off-diagonal blocks change only by a factor of two. This effectively

ties the emissions of $CO_2$ tighter to the emissions of the other species, resulting in more benefit from a multi- over a single-species inversion. Conversely, when all prior uncertainties are reduced by a factor 2 (Case 3), both diagonal and off-diagonal blocks are reduced by a factor four. This explains why Case 1 and Case 3 show similar benefit values. Note that the assumed prior uncertainties for the default case (Case 1) are quite conservative, therefore lower uncertainties were chosen for Cases 2 and 3. While the absolute benefit of adding inter-species correlation is not a game-changer, it is worth pointing out that such

improvement also comes with only slightly greater computational effort than multiple independent single-species inversions.

In order to assess the contribution of inter-species correlation in the prior uncertainty vs. that of model-data mismatch uncertainty, Fig. 11 also shows the resulting posterior uncertainty ratios for Case 1 (Table 3) from inversions only using prior or model-data mismatch correlation. For the anthropogenic component of $CO_2$, the greatest constraint is given by the

prior correlation, while for GEE, respiration, and $CH_4$ the strongest contribution is from the model-data mismatch correlation. In the case of CO, the inter-species correlations for different components are dominant for different months of the year. What makes CO sensitive to different correlation structures during different seasons is that CO enhancement shows a stronger seasonal cycle compared to e.g. fossil fuel component of the $CO_2$ enhancement, with average values for January of around 150 ppb (25 ppm for $CO_2$), and for July of 9 ppb (4 ppm for $CO_2$). This results in a much weaker constraint on the

CO emissions from the CO observations during summer, but still some constraint through the other species such as $CO_2$ via the a priori correlation in the emissions.

Palmer (2006) (in the following referred to as P06) studied the importance of inter-species correlation to improve inverse analysis using airborne data from the TRACE-P mission conducted in March/April 2001 over the western region of the

Pacific Ocean. P06 derived a prior error correlation lower than 0.2 by analysing the uncertainty of emission factors from an Asia-specific emission inventory (Streets, 2003), which is significantly smaller than the correlation of 0.7 assumed in the present study. P06 deemed $CO_2$-CO prior correlation greater than 0.5 to be unrealistic for the emissions in Asia, which is mostly associated with the uncertainty in emission factors for CO of 67% for fossil fuel emissions and 240% for biofuel emissions in China (P06 Table 1). However, for the European region used in the present paper we argue that values around

0.7 are appropriate. The resulting uncertainty in the $CO_2$-CO ratio, diagnosed from the prior error covariance matrix used in this study, is about 50% for both biofuel and fossil fuel emissions in Europe, which we regard as reasonable. To compare results from P06 with those from the present study, ratios of posterior uncertainties resulting from inversions using correlations between $CO_2$ and CO of 0.7 in the prior uncertainties and to those using no correlations have been extracted from Fig. 7 in P06 and are also shown as orange diamonds in Fig. 11. It is easy to see that for anthropogenic $CO_2$, the value derived from P06 is higher than in our study, while the two values are very similar for CO. Similarly, posterior uncertainty ratios using model-data mismatch correlations of 0.7 between $CO_2$ and CO are derived from Fig. 8 of P06 and are shown as red diamonds. In this case, the value derived from P06 is slightly lower than in our study for anthropogenic $CO_2$, while the two are again very similar for CO.

From this comparison we can see that the estimates of the benefit of including inter-species correlation in atmospheric inversions in P06 and in this paper are on the same order of magnitude for anthropogenic $CO_2$ and almost identical for CO, suggesting a general continuity of results.

## 4. Conclusions

The present paper presents a synthetic experiment aiming to evaluate the effects of exploiting correlations between different trace gases in an atmospheric inversion. We quantitatively described the capability of the modeling framework to reproduce observations, the performance of the inversion scheme in reducing the uncertainty of the different trace gases, and the benefits of multi-species inversions compared to corresponding single-species inversions. We also describe a method to re-scale different prior uncertainty covariance matrices so that the corresponding posterior uncertainties are actually comparable.

Where possible, we confronted model outputs with available observations. Such comparison, possible only for CO, showed a good degree of agreement between the model and observations with an overall correlation of roughly 0.75; modeled values for CO enhancement underestimate the observed ones by a factor of roughly 2.8, compatible with what was found in

Boschetti (2015). It is found that posterior uncertainty is much lower than the prior for all of the five simulated species. The mean uncertainty reduction for $CO_2$ emissions from fossil fuels is roughly 38%, for GEE it is around 41% while for respiration it is roughly 44%. For CO and $CH_4$ the uncertainty reduction is about 63% and 67% respectively. Finally, we described quantitatively the benefit of using multi-species inversions by exploiting correlations in different chemical species. It is found that considering correlations between different trace gases can reduce the posterior uncertainty by up to about 20% for monthly fluxes. These benefits are however dependent on the error structure of the prior uncertainty.

The present paper paves the way for future studies using simultaneous measurements of different trace gases. This will be especially important in the context of the upcoming routine measurements of $CO_2$, CO, and $CH_4$ vertical profiles within IAGOS. As IAGOS makes use of commercial airliners, such profiles will be collected in the vicinity of major international airports, and hence in the vicinity of major metropolitan areas, where many different human activities take place simultaneously. In such a context, any improvement in the constraint of atmospheric inversions will be particularly useful. A possible improvement in this analysis would be to evaluate the effects of different correlation factors specific to different pairs of anthropogenic species, fuels and emission sectors.

## 5. Acknowledgements

The research leading to these results has received funding from the European Community's Seventh Framework Programme ([FP7/2007-2013]) under grant agreement n° 312311 (IGAS).
The authors acknowledge the strong support of the European Commission, Airbus, and the Airlines (Lufthansa, Air-France, Austrian, Air Namibia, Cathay Pacific, Iberia and China Airlines so far) who carry the MOZAIC or IAGOS equipment and perform the maintenance since 1994. In its last 10 years of operations MOZAIC has been funded by INSU-CNRS (France), Météo-France, Université Paul Sabatier (Toulouse, France) and Research Center Jülich (FZJ, Jülich, Germany). The IAGOS database is supported by AERIS (CNES and INSU-CNRS). Data are also available via AERIS web site http://www.aeris-data.fr

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

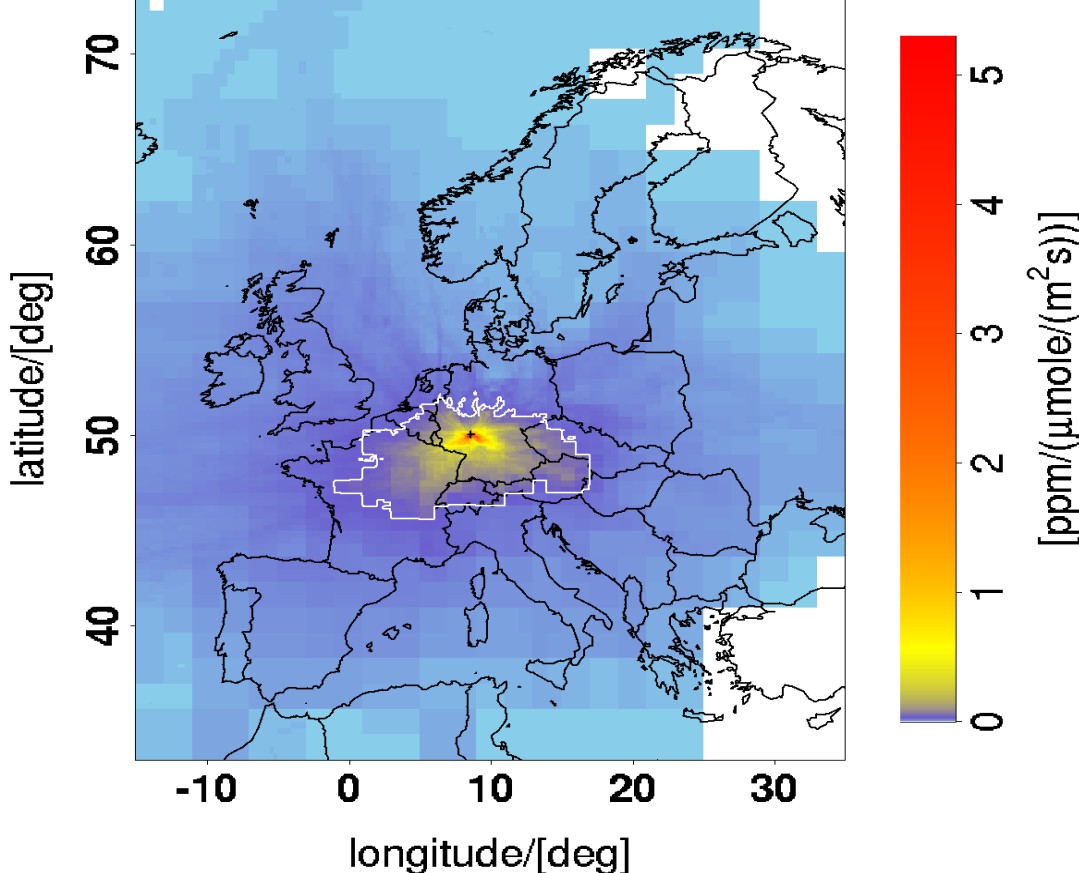

**Figure 1: Cumulative sum of the ML footprints for all the flights into or out of FRA in the year 2011. The gray line delineates the 50% footprint.**

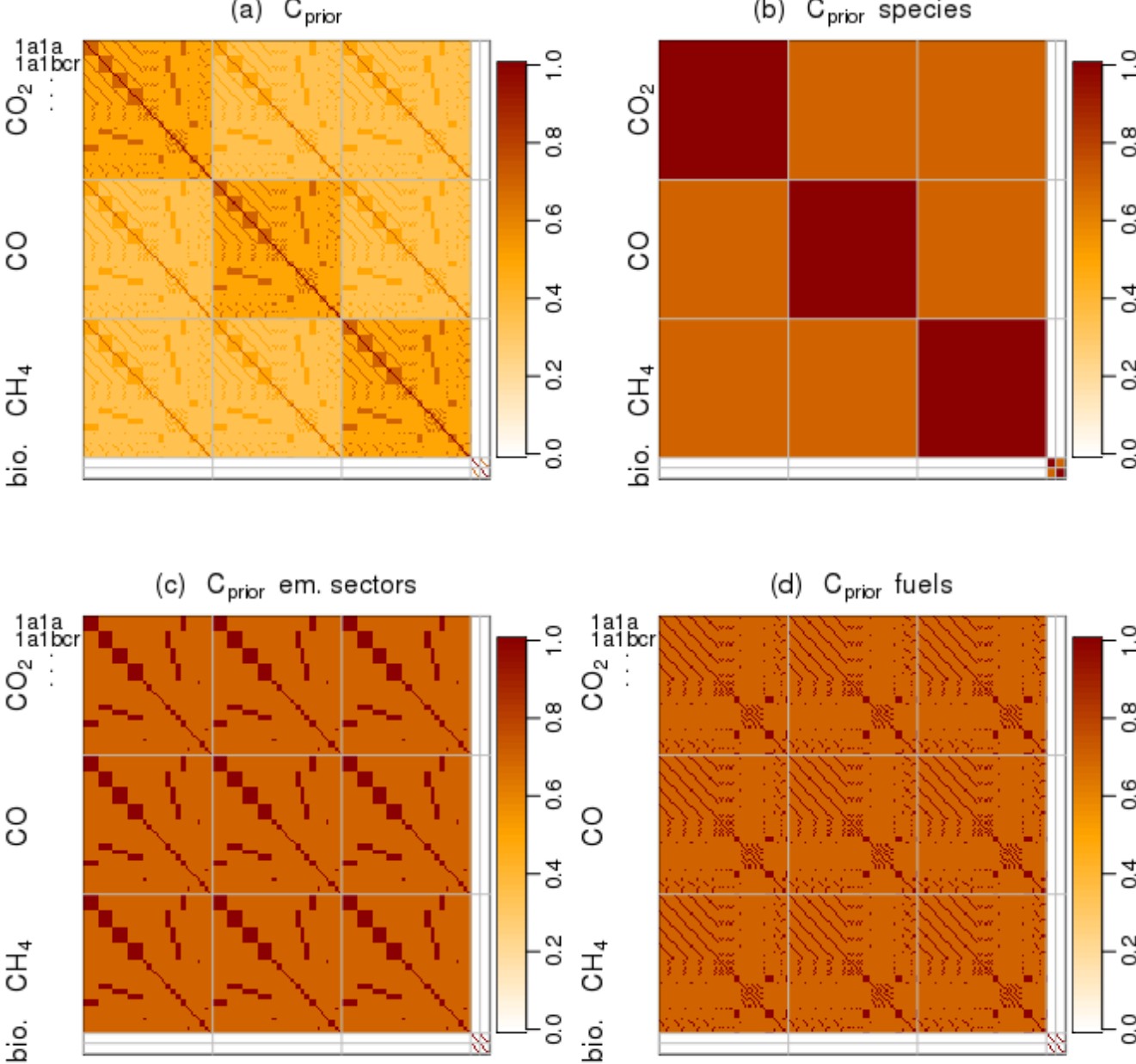

**Figure 2: Prior error correlation matrix (a) used in the multi-species inversion, and the respective components for modeled species (b), emission sectors (c) and fuel types (d). Matrix (a) is the element-wise product of matrices (b), (c) and (d). Each matrix has the same dimensions (2604x2604) reflecting the length of the state vector. The matrices are shown for only one month here, for illustration. The gray lines indicates subsets of the flux categories according to different modeled species ('blocks'), ordered as follows from top to bottom and from left to right: anthropogenic $CO_2$, CO, $CH_4$, GEE and respiration. In the single-species inversion, the correlation values in the off-diagonal 'blocks' of matrix (b) are set to zero. In the complete matrix, correlation between fluxes from different months is also set to zero.**

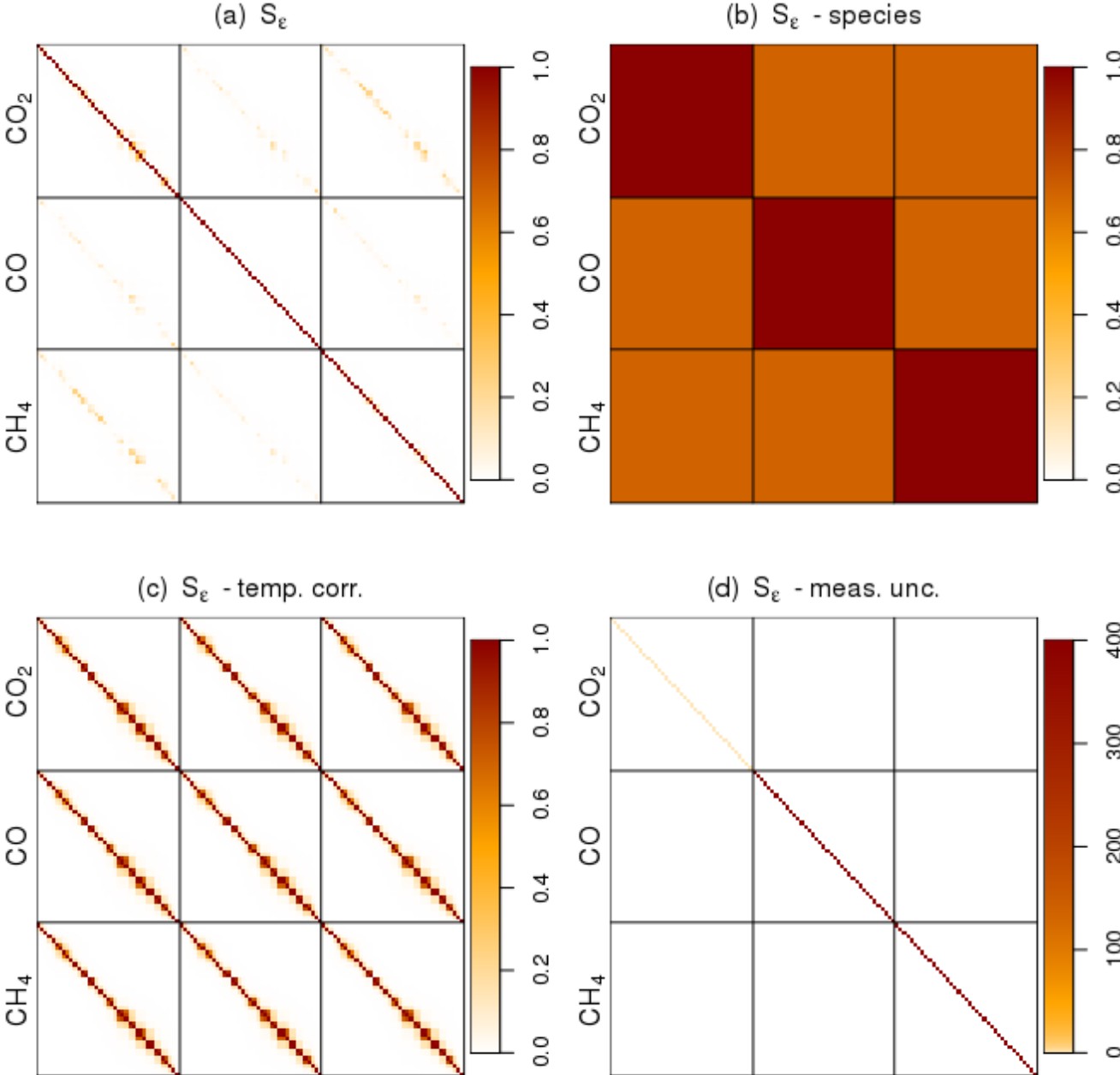

**Figure 3: Model-data mismatch correlation matrix (a) used in the multi-species inversion, species correlation matrix $S_s$ (b), temporal correlation matrix $S_t$ (c) and squared measurement uncertainty (d). Note that the measurement uncertainty is expressed in ppm for $CO_2$ and ppb for CO and $CH_4$. Each matrix has the same dimensions (1098x1098) reflecting the length of the observation vector, but here only the data of July are plotted to increase visibility. The gray lines indicate different species in the observation vector ('blocks'), ordered as follows from top to bottom and from left to right: total $CO_2$, CO and $CH_4$. In the single-species inversion, the correlation value in the off-diagonal 'blocks' of matrix (b) is set to zero. The structure in $S_s$ in (c) is a result of the uneven temporal distribution of the observations within the month.**

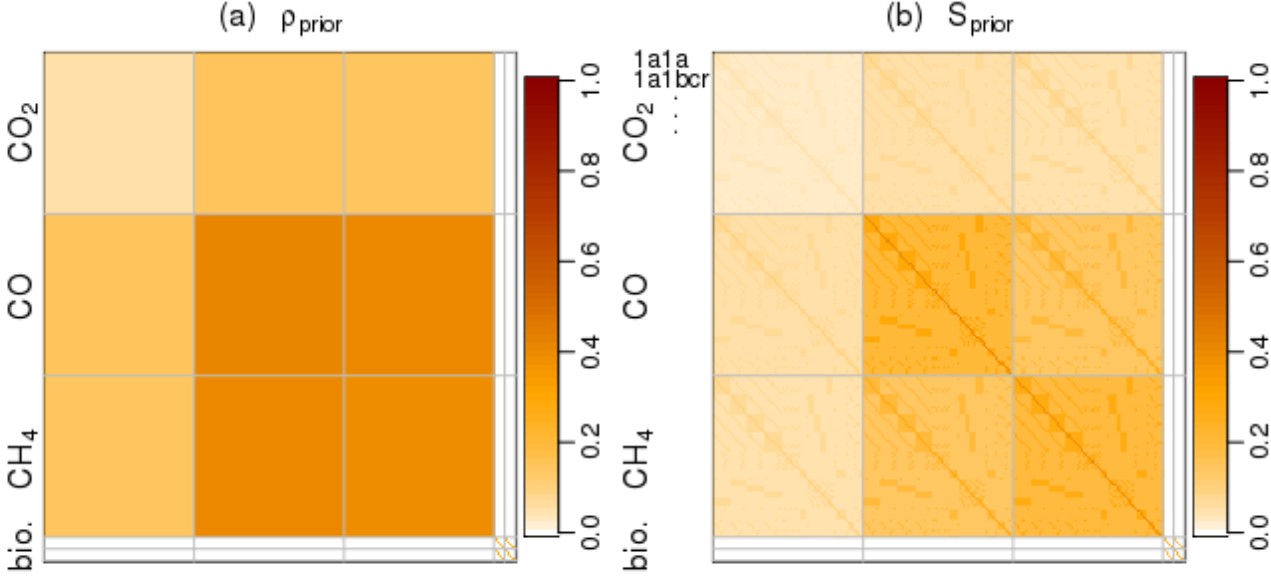

**Figure 4: The final rescaling matrix $\rho_{prior}$ (a) and the prior error covariance matrix $S_{prior}$(b). Note that $\rho_{prior}$ can be defined as the element-wise ratio of $S_{prior}$ and $C_{prior}$.**

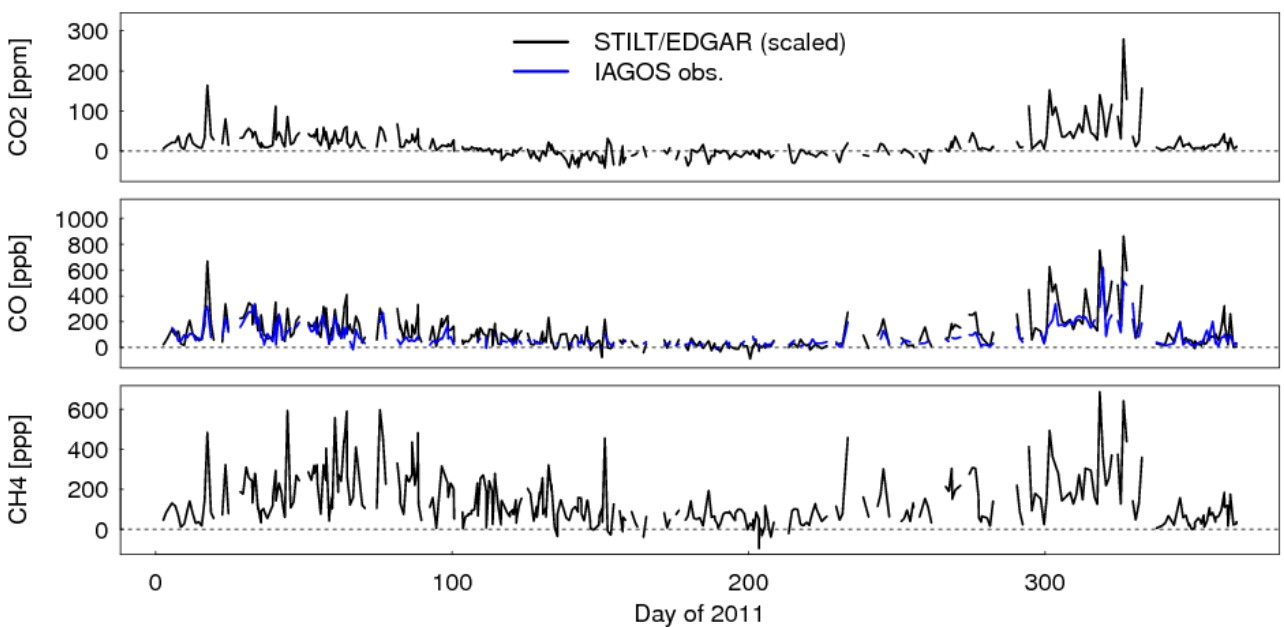

**Figure 5: Mean daily enhancement of mixed layer vs. free tropospheric mole fractions. Modeled mixing ratios are shown as black lines, while the observed CO is shown as blue line. Note that the modeled values for CO have been multiplied by a factor of 2.8, corresponding to the mean ratio between observed and modeled CO enhancements, to match the observed values.**

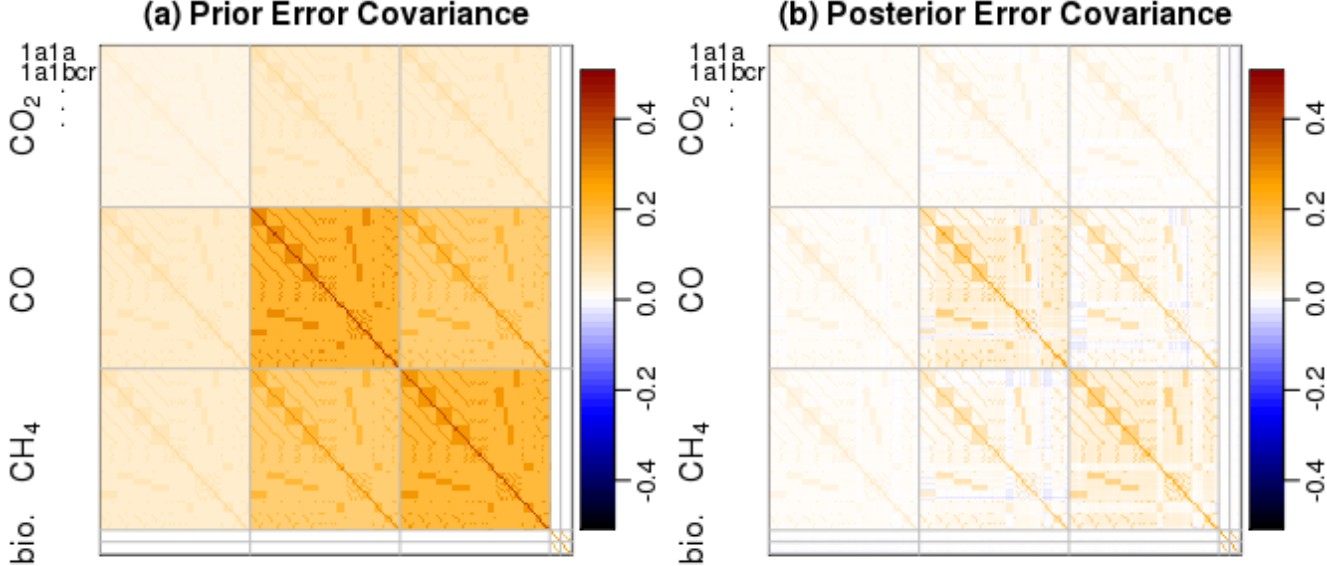

**Figure 6: Prior error covariance matrix (left) and corresponding posterior error covariance matrix (right).**

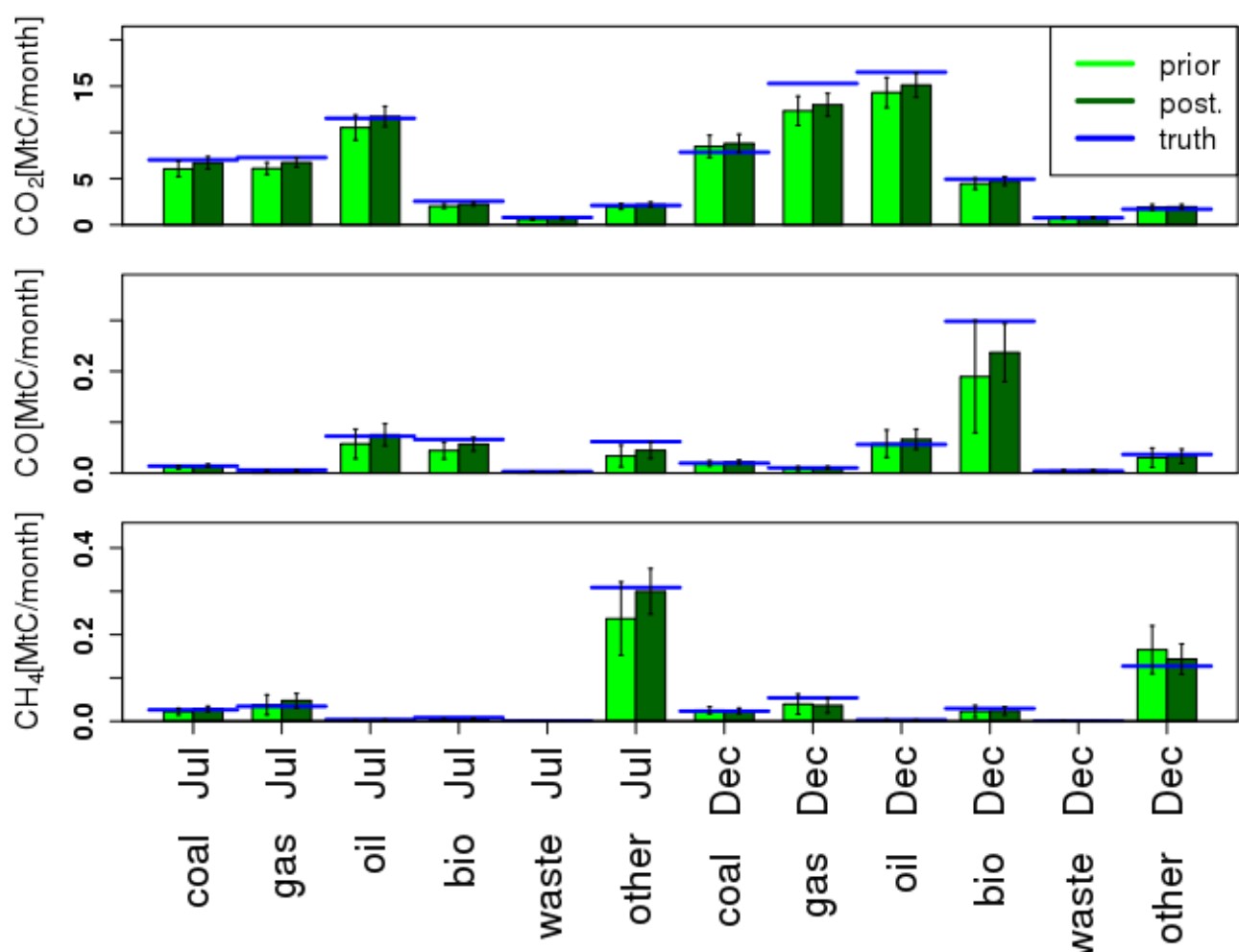

5 **Figure 7: Prior, posterior and true (pseudo-data) fluxes in physical units aggregated for different fuel types. Note that, as the true fluxes are the result of a random perturbation of the prior, they do not describe an actual situation in the physical world. So, for example, the fact that the true value of CH$_4$ fluxes in July is lower than the same value in December should not be surprising.**

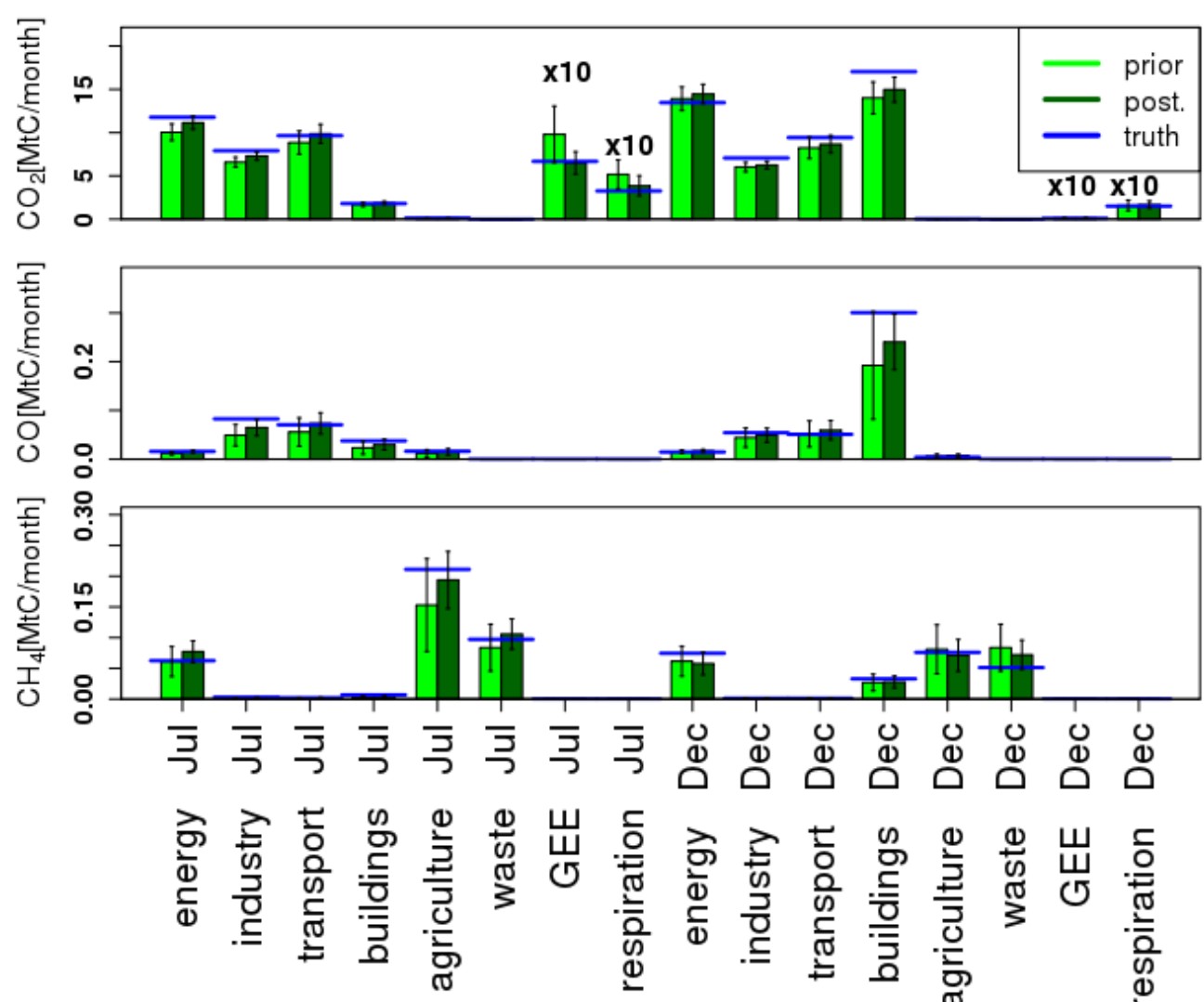

**Figure 8: Prior, posterior and true (pseudo-data) fluxes in physical units aggregated for different emission sectors. Absolute values of biosphere-atmosphere exchange fluxes of $CO_2$ are included in (b), but scaled down by a factor of 10. Note that, as the true fluxes are the result of a random perturbation of the prior, they do not describe an actual situation in the physical world. So, for example, the fact that the true value of CO for transport in July is higher than the same value in December should not be surprising.**

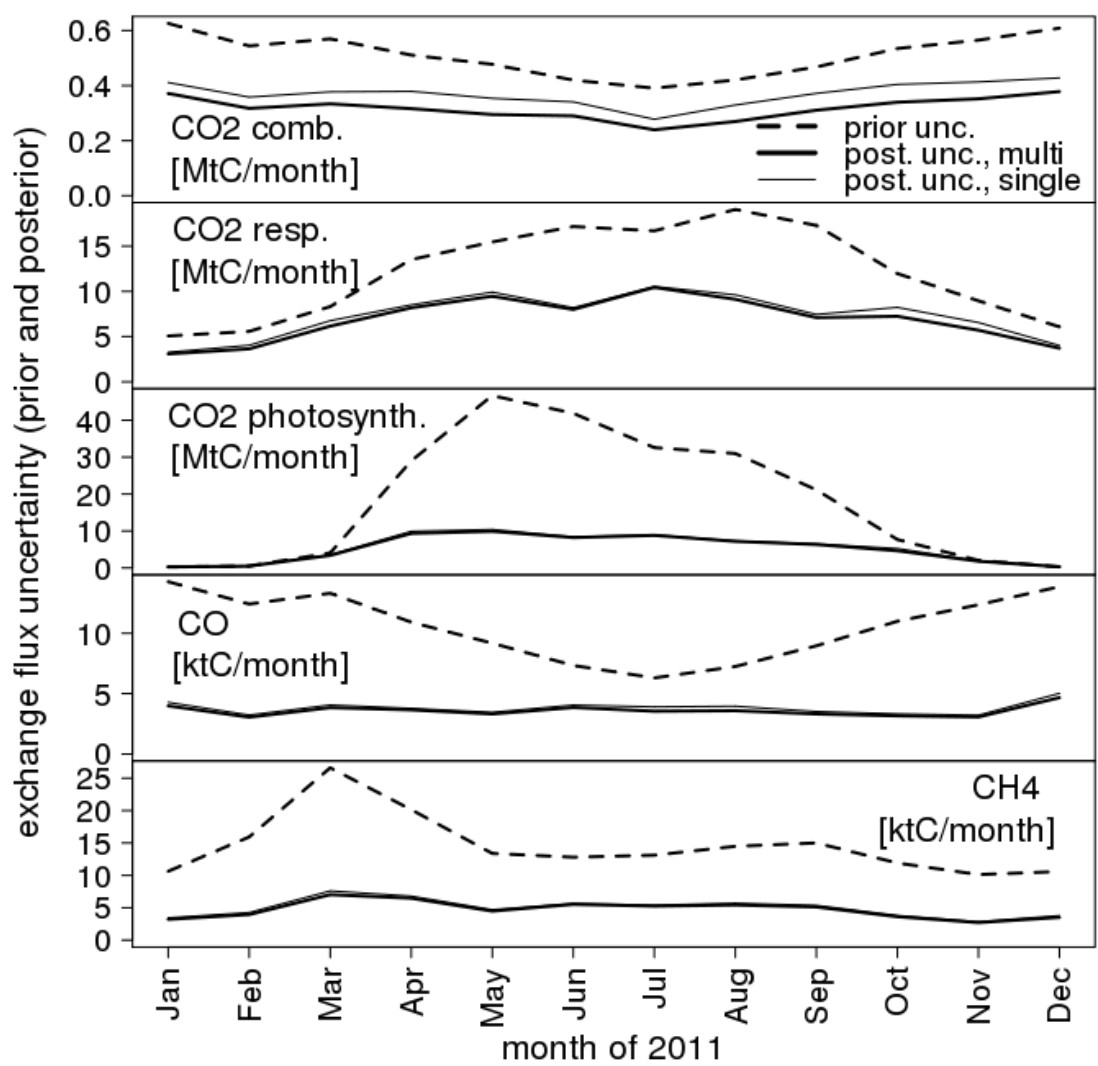

**Figure 9: Comparison between prior and posterior monthly uncertainties for the five tracers. The posterior uncertainty is plotted for both the multi-species inversion, accounting for inter-species correlations, and the single-species inversion, in which all of the species are independent. Both prior and posterior uncertainty are expressed in physical units. The spike in the prior methane uncertainty estimate for the month of March depends on the emission inventory and is related to the cycle of agricultural activities.**

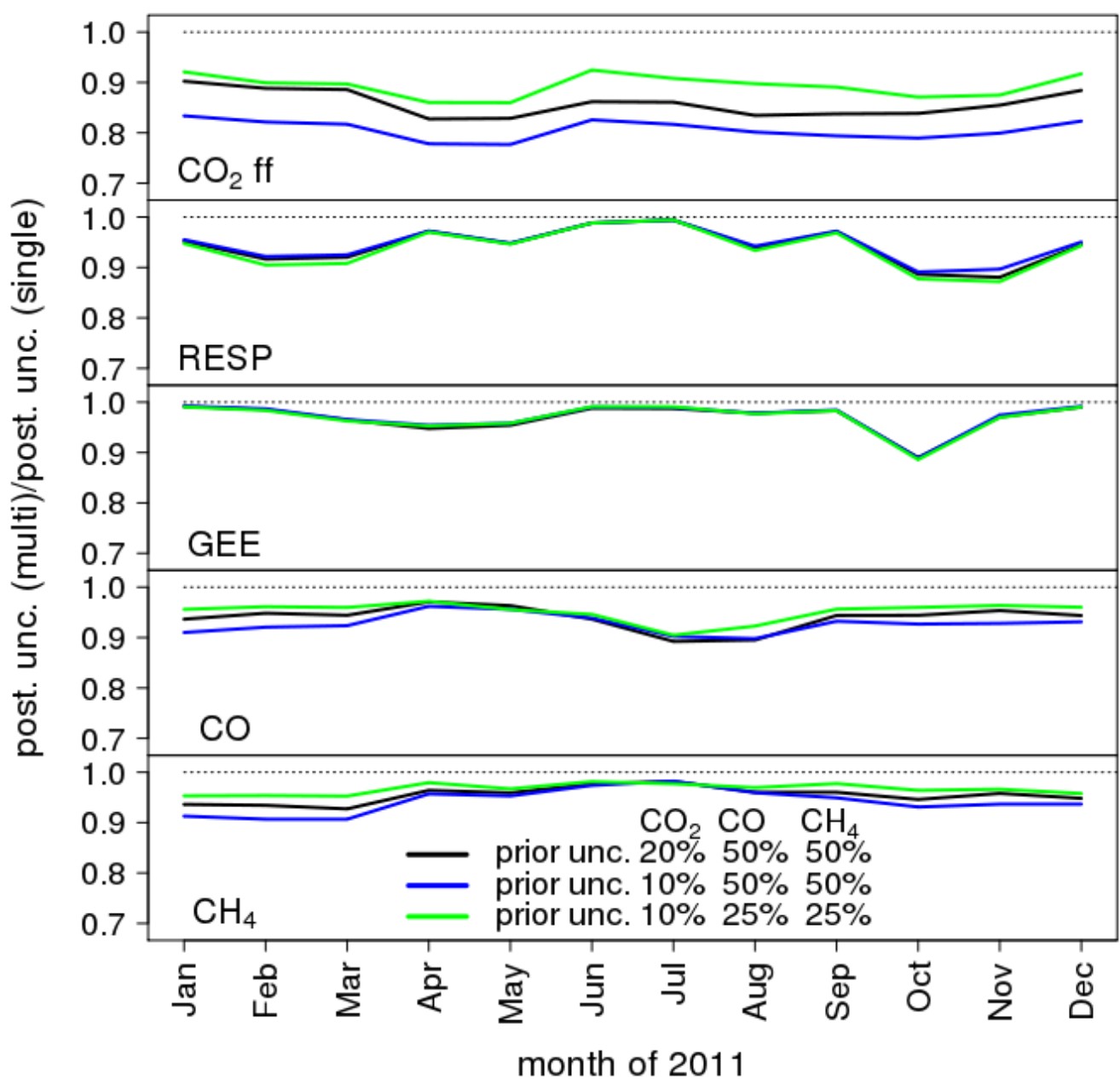

**Figure 10: Benefit of a multi-species inversion over the corresponding single-species (dotted line) per different species per months of the year. The benefit has been tested for the three different cases of Table 3. Note that CO₂ refers to fossil fuel emissions only, and RESP and GEE refers to the biospheric fluxes.**

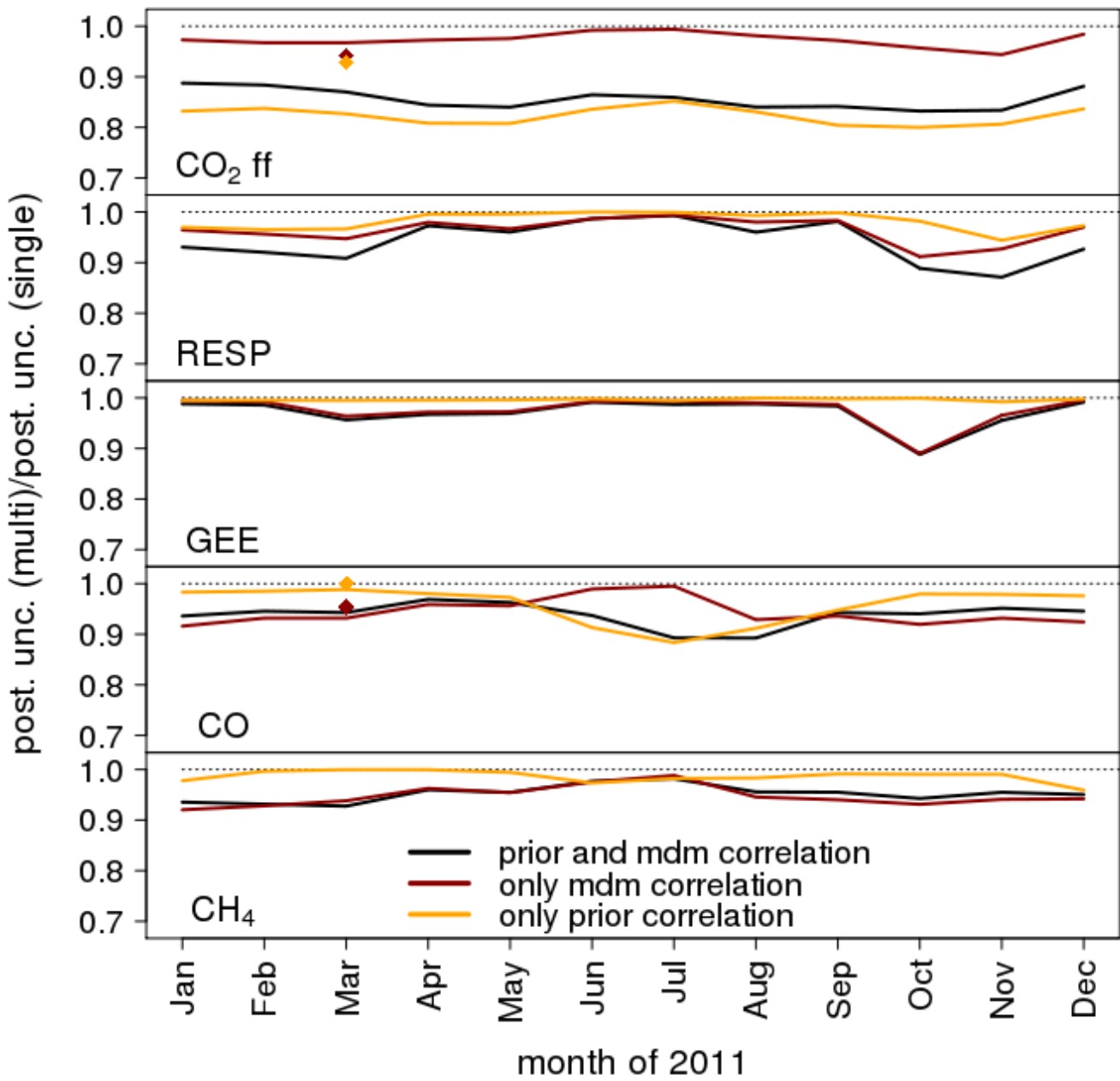

**Figure 11: Benefit of a multi-species inversion over the corresponding single-species (dotted line) per different species and month. The benefit has been tested for a "normal" inversion featuring both prior and model-data mismatch correlation between different species (black) or only one of these two components (red and orange). Results refer to Case 1 of Table 3 (black line of Fig. 10). Values derived from Palmer (2006) for the month of March are indicated with a diamond.**

| | Adj IPCC | Description | Aggregated |
|---|---|---|---|
| 1 | 1a1a | Power generation | Energy |
| 2 | 1a1bcr | Other transformation non-energy use | Energy |
| 3 | 1b1 | Solid fuels production | Energy |
| 4 | 1b2abc | Gas flaring | Energy |
| 5 | 1b2ac | Oil prod., distribution and flaring | Energy |
| 6 | 1b2b | Gas production and distribution | Energy |
| 7 | 1a3a+1c1 | International and domestic aviation | Transport |
| 8 | 1a3b | Road transport | Transport |
| 9 | 1a3ce | Non-road ground transport | Transport |
| 10 | 1a3d+1c2 | Inland waterways and shipping | Transport |
| 11 | 1a2+6cd | Industrial combustion (non-power) | Industry |
| 12 | 2a | Cement and lime production | Industry |
| 13 | 2befg+3 | Chemical industry and solvent | Industry |
| 14 | 2c | Metal industry emission | Industry |
| 15 | 1a4 | Buildings | Buildings |
| 16 | 4a | Enteric fermentation in agriculture | Agriculture |
| 17 | 4b | Manure management | Agriculture |
| 18 | 4c | Rice cultivation | Agriculture |
| 19 | 4f | Agricultural waste burning | Agriculture |
| 20 | 6a | Solid waste disposal in landfills | Waste |
| 21 | 6b | Wastewater treatment | Waste |
| 22 | 7a | Fossil fuel fires | FF_fuels |

**Table 1: Specific emission sectors accounted for in the state vector and aggregated categories as used in Fig. 8.**

| | Fuel type | Aggregated fuel type |
|---|---|---|
| 1 | Brown coal | Coal |
| 2 | Hard coal | Coal |
| 3 | Peat | Coal |
| 4 | Gas derivatives | Gas |
| 5 | Natural gas | Gas |
| 6 | Heavy oil | Oil |
| 7 | Light oil | Oil |
| 8 | Solid waste | Waste |
| 9 | Venting and flaring | Oil |
| 10 | Other (*) | Other |
| 11 | Gas biofuels | Bio |
| 12 | Liquid biofuels | Bio |
| 13 | Solid biofuels | Bio |

**Table 2: Specific fuel types accounted for in the state vector and aggregated categories as used in Fig. 8.**
**(*) The category "Other" is derived by summing the contribution from those processes in which is difficult to establish the specific fuel responsible for the emissions.**

|        | $CO_2$ | CO  | $CH_4$ |
|--------|------|-----|------|
| Case 1 | 20%  | 50% | 50%  |
| Case 2 | 10%  | 50% | 50%  |
| Case 3 | 10%  | 25% | 25%  |

**Table 3: relative uncertainty of the prior fluxes aggregated domain-wide and annual for the different cases**

| | Prior - Truth (MtC y$^{-1}$) | Posterior - Truth (MtC y$^{-1}$) | Pert. Prior - Truth (MtC y$^{-1}$) |
|---|---|---|---|
| $CO_2$ ff | -14.2 | 1.5  (-111 %) | -8.8  (-38 %) |
| CO | -0.95 | -0.29  (-69 %) | -1.08  (+13 %) |
| $CH_4$ | 0.36 | 0.11  (-68 %) | 0.84  (+133 %) |
| GEE | -81.8 | -17.9  (-78 %) | -116.8 (+43 %) |
| Respiration | 39.5 | 20.6  (-48 %) | 62.2  (+58 %) |

**Table 4: Overall bias for different species between the prior and both posterior and perturbed prior. The percentage values in parenthesis refer to the corresponding Prior-Truth bias.**

| Correlation | Post-Truth $CO_2$ ff | Post-Truth CO | Post-Truth $CH_4$ | Post-Truth GEE | Post-Truth Respiration |
|---|---|---|---|---|---|
| 0.1 | -6.3 ±16.4 | -0.3 ± 0.2 | -0.1 ± 0.3 | -18.5 ± 23.6 | -19.0 ±27.5 |
| 0.2 | -4-4 ±16.1 | -0.3 ± 0.2 | 0.0 ± 0.3 | -18.6 ± 23.5 | -19.2 ±27.4 |
| 0.3 | -2.7 ±15.9 | -0.3 ± 0.2 | 0.0 ± 0.3 | -18.6 ± 23.4 | -19.5 ±27.3 |
| 0.4 | -1.3 ±15.6 | -0.3 ± 0.2 | 0.0 ± 0.3 | -18.5 ± 23.4 | -19.7 ±27.3 |
| 0.5 | -0.1 ± 15.2 | -0.3 ± 0.2 | 0.0 ± 0.2 | -18.4 ± 23.3 | -20.0 ±27.2 |
| 0.6 | 0.8 ± 14.6 | 0.3 ± 0.2 | 0.1 ± 0.2 | -18.2 ± 23.2 | -20.3 ±27.1 |
| 0.7 | 1.5 ± 13.7 | -0.3 ± 0.2 | 0.1 ± 0.2 | -17.9 ± 23.2 | -20.6 ±26.9 |
| 0.8 | 1.9 ± 12.4 | -0.3 ± 0.2 | 0.2 ± 0.2 | -17.6 ± 23.1 | -20.9 ±26.8 |
| 0.9 | 1.5 ± 10.4 | -0.4 ± 0.2 | 0.3 ± 0.2 | -17.3 ± 23.0 | -21.1 ±26.5 |

**Table 5: Residuals between total annual posterior fluxes and total annual true fluxes for the five simulated species (in MtC y$^{-1}$) and different inter-species correlation values in the prior error covariance matrix (first column). The corresponding posterior uncertainty was added for each Post-Truth value.**