# Peer review of "Multi-species inversion and IAGOS airborne data for a better constraint of continental scale fluxes"

_Atmospheric Chemistry and Physics, 2017_

## Referee Comment (RC1) · Anonymous Referee #1 · 4 Apr 2017

Overview:

The manuscript "Multi-species inversion and IAGOS airborne data for a better constraint of continental scale fluxes" by Boschetti et al. describes the effect of including the correlations between multiple species in a bayesian inversion framework in order to improve error reduction compared to solving for individual species independently. The experiment described in the manuscript uses synthetic observations based on measurements made during the IAGOS campaign in Europe in order to assess the potential for future measurements of $CO_2$, CO and $CH_4$ during this campaign to better constrain regional emissions of all three species. Finally, there is some discussion of the effect of different assumptions about the prior error of the emissions upon the level of error reduction achieved by the inversion.

Overall the manuscript is fairly well written, with few technical corrections necessary. The figures are generally quite clear and well chosen, although some further detail needs to be provided for some of them. The methods and models used within the manuscript are appropriate for such a study, and are able to provide some assessment of the potential for improvement supplied by future multi-species measurements as part of the measurement campaign. The paper is successful as far as it goes, and whilst it would have been nice to further examine the effect of different experiment set-ups within this paper, the authors acknowledge that this is the case, and may be the focus of a future manuscript.

My main reservation with the study is that the results and discussion section is a little light on detail in places and feels like it was rushed, making the thread of the paper more difficult to follow than it should be. More details and deeper analysis of the results is needed in order to contextualise the findings of the experiment. The authors must make sure that all terms used have been explained or defined, and that they provide enough analysis of their results. See general comments for details. I suggest that this paper is suitable for publication in this journal after the following revisions are carried out and the results section is improved.

Comments:

Page 3 line 6: "Because most biogenic fluxes in Europe are influenced by human activities…" - reference?

Page 4, lines 1-2: "proven to be important in the fields of…" - reference?

Page 8, line 18: the first term in equation (3) should be to the power of (-1).

Page 8, line 31: the term "50% footprint" should be explained.

Page 8, line 15: is it fair to assume no correlation between months? You should comment here (or later in the discussion) on whether this would be the best set-up of the correlation matrix in an inversion using real observational data.

Page 12, line 4: What is *enh*?

Page 12, line 7-8: You need to explain how you derive $\varepsilon_{tran\_v}$ in more detail here.

Page 12, line 18: What method do you use to invert $\mathbf{S_{prior}}$ and $\mathbf{S_e}$ ?

Page 13, line 5: Describe which version of the model output you are plotting in Figure 5. Does it use the prior emissions?

Page 13, line 9: Here, and in the caption of Figure 5, you say that the modelled CO is multiplied by a factor of 2.8. However, the legend of Figure 5 appears to say that the observations have been scaled. Which is correct?

Page 13, line 12: Explain here what it is that is indicated by the performance of the model compared to the observations. Are you saying that the meteorology that you use and the correction to $z_i$ that you apply produce a good indication of the temporal variation of the ML enhancement? Does your choice of $z_i$ display an improvement over the original?

Page 13, lines 25 and 26: You could probably add a little more detail to this one-sentence paragraph. Explain that Figure 6 is showing the prior and posterior emission error covariance matrices for the base multi-species inversion. Do the single-species matrices show a similar overall error reduction? Do you expect to see negative correlations in the posterior matrix? As it stands this sentence is disjointed and appears to come out of nowhere and doesn't relate to other text, making the manuscript unnecessarily difficult to follow.

Page 14, lines 24 - 28: Explain what you mean by "a perturbed version of the prior" here. Also, does the multi-species inversion capture the "truth" any better or worse than the single-species inversion?

Page 15, line 19: How robust do you think the relative uncertainty reductions that you derive are against different manifestations of the "true" fluxes?

Page 15, line 26: Why do you think a smaller prior error for the CO2 FF fluxes compared to the other species leads to a greater uncertainty reduction for the posterior fluxes?

Page 16, line 3: What makes CO sensitive to different correlation structures during different seasons?
Technical corrections:

Page 1, line 13: no comma needed in "for, GEE"

Page 1, lines 17 and 18: the percentages reported in the abstract here are in some cases slightly different to those reported in the main text of the manuscript (on page 15).

Page 2, line 2: difference -> differences

Page 5, line 10: Matherial -> Material

Page 10, line 3: Section 2.1.6 -> Section 2.1.5

Page 16, line 18: Delete "meaning" - or explain what it means.

---

## Referee Comment (RC2) · Anonymous Referee #2 · 28 Jun 2017

The authors present a multi-species inversion system for inferring CO, CO2 and CH4 fluxes from future IAGOS measurements. Their OSSE experiments show that the mutli-species inversion with assumed prior (and observation) error correlations can improve top-down flux estimate, resulting in a smaller posterior uncertainty than the simple approach where prior errors are assumed to be independent. Overall the manuscript is very informative, and their results are meaningful. It can be accepted for publication in ACP after minor revision.

Major comments: 1. While I agree that inclusion of prior error correlations between different emissions can improve observation constraint, and help disentangle sources, improper characterization of the error correlation may result in systematic bias in the posterior estimate. So I suggest a more complicated OSSE is necessary, where per-

turbations are generated using different correlation parameters to exam how well the system will reproduce the 'true' fluxes, with incorrect correlation coefficients.

2. Discussions are more focused on the domain total. It is interesting to see how well the system will reproduce their spatial distribution.

Minor comments:

1. Line 5, Page 4: "This synergy follows from the fact . . ." Better changed to 'follows the fact. . .' or other phrase.

2. Line 5, Page 14: ". . .have a magnitude of 6-11 Megatons of carbon per year (MtC y-1) in July"

The unit of MtC/yr seems inconsistent with annual total presented in Table 5. I think it should be MtC/a.

3. Line 27, page 14: ".. .for the whole year between the prior and both posterior and the perturbed prior" The sentence is unclear.

4. Figure 1: Please explain why for CH4 fluxes in December, their uncertainty has been significantly reduced, but the differences from the 'true' are not obviously improved.

5. Figure 6: I suggest the authors also provide the prior and posterior error correlation between CO2 fossilfuel emission and biospheric net flux in the main text.

6. Figure 7: check the units for monthly fluxes (in main text as well).

———————————

---

## Referee Comment (RC3) · Anonymous Referee #3 · 12 Jul 2017

General comments:

The paper presents a multi-species inversion framework tested using pseudo-data experiments. Various assumptions are made to evaluate the sensitivity of the inversion, with an emphasis on the impact of error correlations across species and sectors. Overall, the paper presents an innovative approach to assimilate various atmospheric species in a single inversion framework. This study is clearly worthwhile publishing but lacks a better evaluation of the aggregation operator assumption (perfect prior emission distribution) and the impact of systematic errors in the system affecting the correlations in the gas-sector attribution problem. The Observing Ssystem Simulation Experiments (OSSE's) cover some of the assumptions with varying levels of uncertainties but several components are not carefully considered. The two major concerns here are the

aggregation operator, that remains perfectly known and so the spatial distribution of the prior fluxes, and the assessment of correlations among sectors and across trace gases for the different species that remain very unclear. A last but less critical concern is related to the assumption that transport errors are similar across species, which is unlikely for CH4 and CO2 for example, rarely co-emitted (only CO2-CO is discussed) and therefore affected by different problems in different parts of the domain. The work focuses primarily on random errors and ignores systematic errors that remain the main limitations in atmospheric inversions. Therefore, this study requires some additional experiments before publication, specifically addressing the error associated with the aggregation operator and errors in gas ratios for the different sectors.

- The use of an aggregation operator needs to be discussed. Hyper-parameters (here scaling factors for the sectors) are used to reduce the dimension of the problem but corresponds to an assumption of perfectly-known distributions. The system should be evaluated not only under the "perfect spatial distribution" assumption, especially for CO2 biogenic fluxes which are clearly not well-known. One suggestion to clarify the concern here would be to use VPRM as truth but assumes a different distribution when constructing the aggregated solution such as the posterior fluxes from Panagiotis et al. (2016). Other experiments could be designed here to test the aggregation problem. Similarly, the area defined by half of the total footprint is arbitrary and never tested nor justified. Why 50% was used? How much variations are expected within that area which would affect the error correlations? If a power plant is located near an airport, how would that affect the CO/CO2 correlations and therefore the homogeneity within the aggregated area?

- The discussion about error correlations across species is confusing. How did you define the emissions for the different sectors? Have you assigned gas ratios to various sectors? If so, what are these ratios? Some of the discussions are related to using CO2 and CO data to diagnose gas-to-gas correlations, but the exact definition of the emissions of the different gases for each sector has been defined in the inversion

system. Or maybe the sectors are unrelated for each gas? The different sectors have ratios int erms of trace gas emissions but these emission ratios vary regionally. This section needs to be explained in more details. The assumptions made here should also be tested in the inversion framework.

- CO biogenic fluxes: the paper does not address the problem of CO biogenic fluxes during the growing season. Warm days in summer correspond to large amount of biogenic VOC's beign emitted from the vegetation, producing CO to non-negligible levels. This issue should be discussed if not addressed. How would this problem affect the ability to retrieve the truth?

- When you constructed your error correlations for CH4, transport errors are unlikely to be highly correlated as CH4 is only partially co-emitted with CO2 and CO. Large emissions from NG production and farming activities are uncorrelated with biogenic or fossil fuel consumption. This problem should be adressed here. If transport errors, which are spatially variable, affect CH4 and CO2/CO in different ways, the error correlation would be affected. Additional experiments using incorrect error correlations would quantify the sensitivity of the inverse fluxes to the assumptions made in prior errors.

- The problem of unreported sources in CH4 inventory is not addressed at all. Recent papers have discussed the lack of information for natural gas and oil production operations, or from recent and old mining areas. How would unreported sources affect the inverse solutions? This question comes back to the aggregation operator.

- The utility of the figures showing the multiple error covariance matrices for the different cases remains limited. The information content would be better described with words or mathematically. Readers cannot extract useful information from contour plots of covariance matrices. They could remain part of the paper but as part of the supplementary information. A table could also synthesize the various assumptions tested in the inversion system.

Technical comments:

3-1: Consequently, intercomparisons...

3-3: the international level

3- 1st paragraph: This paragraph is confusing and not always following a logical path. Prediction skills and emission reduction are two different problems not directly connected to each other. Explain better the broad context of this study by focusing on the main general issues and clarify which one you are trying to address here.

3-10: A commonly used approach to estimate...

3-13: Actually, the uncertainty reduction relies purely on the assumptions made in the system and not on the effective ability of the system to produce a reliable solution. Bayesian system assumes that data will improve the a priori by construction. Explain better what you mean here.

3- 2nd paragraph: Several papers are missing here. For example, CO2-CH4 inversion using satellite data (Pandley et al., 2015) or the optimization of co-emitted species (Brioude et al., 2012), and early work on delta 13-CO2 by Enting et al. (1995). THe authors should dig into atmospheric chemistry studies where several studies have addressed the use of multiple co-emitted species to constrain emissions at small scales.

Previous studies using multiple species to constrain emissions should be introduced here, even without having used a formal inversion framework, such as urban studies over Los Angeles (e.g. Peischl et al., 2013). The optimization problem is equivalent and relies on similar ideas to constrain the emissions.

5-24: This technique assumpes that the wind direction and speed are comparable near the surface and at 2km high. Mass-balance studies have shown that this is often not the case (e.g. Karion et al., 2015). Free troposheric air represents different air masses due to the wind direction and speed gradients in the vertical. This assumption would need to be tested with the particle model.

7-3: What about CO biogenic fluxes? During warm summer times, biogenic CO fluxes

represent a significant fraction of the signals. Did you ignore this contribution in your study?

2.1.3 To reduce the dimension of the state vector, you assume here that the spatial distribution of the prior fluxes and emissions are perfect, using an aggregation operator. This approach is reasonable for fossil fuel emissions but less convincing for biogenic fluxes.

12-21: How did you take into acount the truncation of the prior errors? Did you adjust the truncated random perturbations to match the non-truncated assumption made in the prior error covariance matrix? 14-13: The expression "lion's share" should be avoided. A fraction of the contribution is a better metric to describe the importance of the sector.

---

## Author Comment (AC1) · 15 Dec 2017

Dear Referee #1
Thank you for thoroughly reading and commenting the manuscript. Please find below the replies to your suggestions; each of your suggestions is followed by the corresponding reply in bold letters and (where appropriate) actions taken to address it in the updated version of the manuscript in italics.

Comments:

The manuscript "Multi-species inversion and IAGOS airborne data for a better constraint of continental scale fluxes" by Boschetti et al. describes the effect of including the correlations between multiple species in a bayesian inversion framework in order to improve error reduction compared to solving for individual species independently. The experiment described in the manuscript uses synthetic observations based on measurements made during the IAGOS campaign in Europe in order to assess the potential for future measurements of $CO_2$, CO and $CH_4$ during this campaign to better constrain regional emissions of all three species. Finally, there is some discussion of the effect of different assumptions about the prior error of the emissions upon the level of error reduction achieved by the inversion.

Overall the manuscript is fairly well written, with few technical corrections necessary. The figures are generally quite clear and well chosen, although some further detail needs to be provided for some of them. The methods and models used within the manuscript are appropriate for such a study, and are able to provide some assessment of the potential for improvement supplied by future multi-species measurements as part of the measurement campaign. The paper is successful as far as it goes, and whilst it would have been nice to further examine the effect of different experiment set-ups within this paper, the authors acknowledge that this is the case, and may be the focus of a future manuscript.

**We appreciate these positive remarks.**

My main reservation with the study is that the results and discussion section is a little light on detail in places and feels like it was rushed, making the thread of the paper more difficult to follow than it should be. More details and deeper analysis of the results is needed in order to contextualize the findings of the experiment. The authors must make sure that all terms used have been explained or defined, and that they provide enough analysis of their results. See general comments for details. I suggest that this paper is suitable for publication in this journal after the following revisions are carried out and the results section is improved.

**Thank you for the constructive comments. In the revised version ...**

Page 3 line 6: "Because most biogenic fluxes in Europe are influenced by human activities..." - reference?
**We have added references and modified the sentence to:**

*"Because most biogenic fluxes in Europe are influenced by human activities, with 22% of Europe's land is dedicated to agriculture (FAO, 2013) and 45 % covered by forests, of which 80% are managed for wood supply (UNECE, FAO, 2011), understanding and managing these biogenic fluxes must also be a component of any policy to reduce anthropogenic emissions."*

Page 4, lines 1-2: "proven to be important in the fields of..." - reference?
**Two references were added; one for IAGOS and one for CONTRAIL (Zbinden et al, 2013; Sawa et al., 2012)**

Page 8, line 18: the first term in equation (3) should be to the power of (-1).
**The equation (3) was corrected accordingly**

Page 8, line 31: the term "50% footprint" should be explained.
**A reference to section 2.1.2 was added to remind the reader of the 'footprint'; the section now reads:**
*"As a spatial aggregation scale we chose an area from which fluxes have a significant contribution to the observations made at Frankfurt. For this we compute the temporally accumulated footprint values for the whole year 2011, and select those spatial pixels that correspond to 50% of the total (spatially integrated) footprint (Fig. 1)."*

**We also modified Section 2.1.2 (Pag. 6, Line 12) to better explain the concept of footprint:**
*"...so-called "footprints". Briefly, for each measurement location and time (also called receptor point), the model releases an ensemble of virtual particles that are driven back in time using simulated wind fields from ECMWF and turbulence as stochastic process; the residence time within the lower half of the mixed layer is used to determine the potential contribution from surface fluxes, and the cumulative sum of these contributions determines the footprint, that identifies the part of the domain with a certain influence on a single receptor point. This footprint is then matrix-multiplied with an emission map to derive the corresponding simulated mixing ratio in a given receptor point."*

Page 8, line 15: is it fair to assume no correlation between months? You should comment here (or later in the discussion) on whether this would be the best set-up of the correlation matrix in an inversion using real observational data.
**In page 10, line 16, the following was added:**
*In this study, we assume a certain annual total domain wide flux uncertainty, and then break it down by sectors, fuels, and months by inflating the error. By assuming no correlation between different months we ensure maximum flexibility in the system to retrieve month-to-month changes based on the observations. Assuming correlation between months would be possible, but has not been investigated here. It is unclear how good the seasonal variation in emissions from the inventories actually is, so in order to not rely too much on these we chose zero correlation.*

*Investigating the effects of different correlation set-ups for the seasonal cycle could be the focus of future research.*

Page 12, line 4: What is *enh*?
**Right after equation (11), the line** *"...where enh indicates the modelled enhancement, and both the horizontal ..."* **was added for clarity**

Page 12, line 7-8: You need to explain how you derive $\varepsilon$ tran_v in more detail here.
**The text (from Page 12, line 6) was edited as follows:**
*... where both the horizontal transport error $\varepsilon_{tran\_h}$ and the vertical transport error $\varepsilon_{tran\_v}$ are characterized as percentage error; $\varepsilon_{tran\_h}$ is assumed to be 50% while $\varepsilon_{tran\_v}$ is profile-specific with mean value about 10%.*
*"The vertical transport error accounts for the fact that the shallower the mixed layer is, the more difficult it is to model the atmosphere. We assume that after $z_i$-correction the remaining error is on the order of 50 m (related to the vertical resolution of the profile data), so the relative error $\varepsilon_{tran\_v}$ is assumed as the ratio of 50 m to the modeled $z_i$; in this way we obtain an error that gets larger the shallower the mixed layer is. "*

Page 12, line 18: What method do you use to invert **Sprior** and **Se** ?
**We assume this comment refers to Pag 8, eq. (3) and (4). The error correlation matrices are inverted using the R-function "solve" of the base package. At pag. 8, line 26 the following was added:**
*In this study, the inverse of the matrix was calculated using the R-function 'solve' from the base package.*

Page 13, line 5: Describe which version of the model output you are plotting in Figure 5. Does it use the prior emissions?
**We edited the text at lines 3-4 as follows:**
*Figure 5 shows ... for both observations and model outputs using prior emissions.*

Page 13, line 9: Here, and in the caption of Figure 5, you say that the modelled CO is multiplied by a factor of 2.8. However, the legend of Figure 5 appears to say that the observations have been scaled. Which is correct?
**The text and the caption were correct. The legend has been corrected accordingly**

Page 13, line 12: Explain here what it is that is indicated by the performance of the model compared to the observations. Are you saying that the meteorology that you use and the correction to $z_i$ that you apply produce a good indication of the temporal variation of the ML enhancement? Does your choice of $z_i$ display an improvement over the original?

**Thank you for pointing this out. We have now investigated the improvement brought about by using the zi correction. The text in line 10 and following were edited as follows:**

*"Mixing ratios are highly variable, but the model produces a good indication of the temporal variation of the ML enhancement; the squared correlation coefficient between observed and modeled CO enhancements is 0.62, while the standard deviation of corrected model and observation residuals is 85 ppb; note that by not accounting for the $z_i$ correction, such values would be 0.56 and 87 ppb respectively. The median of the mixing ratio enhancement for the three trace gases is 2.8 ppm for $CO_2$, 18.6 ppb for CO and 26.6 ppb for $CH_4$."*

**-- Note: we found that in the uploaded version of the paper the zi correction was actually switched off. After switching the correction on, only Figure 5 is affected. The mean uncertainty reduction values are now 35% for CO2_ff, 48% for CO and CH4, 60% for GEE and 63% for respiration. We deeply apologize for the mistake --**

Page 13, lines 25 and 26: You could probably add a little more detail to this one-sentence paragraph. Explain that Figure 6 is showing the prior and posterior emission error covariance matrices for the base multi-species inversion. Do the single-species matrices show a similar overall error reduction? Do you expect to see negative correlations in the posterior matrix? As it stands this sentence is disjointed and appears to come out of nowhere and doesn't relate to other text, making the manuscript unnecessarily difficult to follow.

**We propose to replace the one-sentence paragraph with the following:**

*"Figure 6 shows the prior and posterior error covariance matrices for the base multi-species inversion. The posterior error covariance matrix for the multi-species inversion (Fig. 6b) shows lower values corresponding to an average uncertainty reduction of 23% across all state vector elements, while the posterior error covariance matrix for the single-species inversion (not shown) is characterized by a mean uncertainty reduction of 20%. This result implies that the multi-species inversion improves the uncertainty reduction by roughly 15%. Negative values in the posterior error correlation matrix are to be expected because different categories are bind together by correlations and therefore are not free to vary independently."*

Page 14, lines 24 - 28: Explain what you mean by "a perturbed version of the prior" here. Also, does the multi-species inversion capture the "truth" any better or worse than the single-species inversion?

**We propose to add the two following sentences at Line 25:**

*"Such perturbed version is obtained by adding realization of the prior error to the prior state space, similarly to how the "truth" is obtained. In addition, it was found that the truth-posterior bias of the multi-species inversion is always lower compared to the single-species inversion. Such difference is between 1.7% and 5.7%, according to the simulated species, with an overall value of 2.4%."*

Page 15, line 19: How robust do you think the relative uncertainty reductions

that you derive are against different manifestations of the "true" fluxes?
**What we investigate in Fig. 10 is not the uncertainty reduction, but the benefit from a multi-species inversion over a single-species one.**
**The following text was added at line 19:**
*The benefit of including inter-species correlations shown in Fig. 10 does not depend on different manifestations of the true fluxes, but only on the posterior uncertainty of the multi- and single-species inversions.*

Page 15, line 26: Why do you think a smaller prior error for the CO2 FF fluxes compared to the other species leads to a greater uncertainty reduction for the posterior fluxes?
**What we investigate in Fig. 10 is not the uncertainty reduction, but the benefit from a multi-species inversion over a single-species one. Uncertainty reduction for CO2 FF is actually greater in Case 1 (36%) compared with the other two cases (29% and 21% respectively), as in those cases (2 and 3) the prior is assumed to be known better already.**

**We have stated in the paper (page 15, line 26) that the benefit from a multi-species over a single-species inversion increases, when changing the prior uncertainty for CO2 emissions. We think that the reason is the following: changing the prior uncertainty in CO2 emissions means changing also the off-diagonal blocks linking the different species together (see Eq. 8). However, the diagonal block for CO2 in the prior uncertainty changes by a factor four in that case, while the off-diagonal blocks change only by a factor of two. This effectively ties the emissions of CO2 tighter to the emissions of the other species, resulting in more benefit from a multi- over a single-species inversion. Note that this is related to the required rescaling of the prior error covariance matrix described in section 2.1.5.**

**We suggest adding the following text at line 28:**
*...for this increase in benefit. The reason for both of these results is probably to be searched in Eq. 8. In fact, changing the prior uncertainty in $CO_2$ emissions means to also change the off-diagonal blocks linking the different species together. However, by reducing the anthropogenic $CO_2$ uncertainty from 20% to 10% (Case 2), the diagonal block for $CO_2$ in the prior uncertainty changes by a factor four, while the off-diagonal blocks change only by a factor of two. This effectively ties the emissions of $CO_2$ tighter to the emissions of the other species, resulting in more benefit from a multi- over a single-species inversion. Conversely, when all prior uncertainties are reduced by a factor 2 (Case 3), both diagonal and off-diagonal blocks are reduced by a factor four. This explains why Case 1 and Case 3 show similar benefit values.*

Page 16, line 3: What makes CO sensitive to different correlation structures during different seasons?
**To explain the issue, we added a couple of sentences at line 4:**
*What makes CO sensitive to different correlation structures during different seasons is that CO enhancement shows a stronger seasonal cycle compared to e.g.*

*fossil fuel component of the CO₂ enhancement, with average values for January of around 150 ppb (25 ppm for CO₂), and for July of 9 ppb (4 ppm for CO₂). This results in a much weaker constraint on the CO emissions from the CO observations during summer, but still some constraint through the other species such as CO₂ via the a priori correlation in the emissions.*

Technical corrections:

Page 1, line 13: no comma needed in "for, GEE"
**The text was edited according to the suggestion**

Page 1, lines 17 and 18: the percentages reported in the abstract here are in some cases slightly different to those reported in the main text of the manuscript (on page 15).
**The percentage values were checked and replaced where needed**

Page 2, line 2: difference -> differences
**The text was edited according to the suggestion**

Page 5, line 10: Matherial -> Material
**The text was edited according to the suggestion**

Page 10, line 3: Section 2.1.6 -> Section 2.1.5
**The text was edited according to the suggestion**

Page 16, line 18: Delete "meaning" - or explain what it means.
**The word "meaning" was removed as suggested**

---

## Author Comment (AC2) · 15 Dec 2017

Dear Referee #2
Thank you for thoroughly reading and commenting the manuscript. Please find below the replies to your suggestions; each of your suggestions is followed by the corresponding reply in bold letters and (where appropriate) actions taken to address it in the updated version of the manuscript in italics.

Major comments:
1. While I agree that inclusion of prior error correlations between different emissions can improve observation constraint, and help disentangle sources, improper characterization of the error correlation may result in systematic bias in the posterior estimate. So I suggest a more complicated OSSE is necessary, where perturbations are generated using different correlation parameters to exam how well the system will reproduce the 'true' fluxes, with incorrect correlation coefficients.
**This is a very useful suggestion, which we followed now. We propose to add the following at Page 14, Line 29**
*"Improper characterization of the error correlation may result in systematic bias in the posterior estimate. As mentioned in Sect. 2.1.6, inter-species correlation, the correlation between different fuel types and the correlation between different emission sectors in **Sprior** is assumed equal to 0.7 (Sect. 2.1.4). To assess how well the system will reproduce the 'true' fluxes with incorrectly specified correlations, a series of experiment was performed in which the inter-species correlation in **Sepsilon** remains equal to 0.7, while the three correlation coefficients in **Sprior** assume different values ranging from 0.1 to 0.9. Table 5 shows the residuals between total annual posterior fluxes and total annual true fluxes for the five simulated species, derived similarly as for Table 4.*
*We found that for all species the uncertainty reduction increases with correlation. In addition, the posterior-truth biases are always lower than the prior-truth biases.*
*The posterior uncertainty values are almost always larger then the corresponding bias values, except for CO with prior correlation equal 0.8, and fossil fuels $CO_2$ with prior correlations equal to 0.6, 0.7 and 0.9. Follows that, except for these few cases, the posterior is not significantly different from the truth. Conversely, prior-truth biases (not shown) are statistically significant in the majority of cases for fossil fuel fluxes, and in some cases also for biogenic fluxes. The effect of assuming the incorrect error correlations appears to be in general small, possibly implying a relative robustness of our methods.*

| Correlation | Post-Truth $CO_2$ ff | Post-Truth CO | Post-Truth $CH_4$ | Post-Truth GEE | Post-Truth Respiration |
|---|---|---|---|---|---|
| 0.1 | -17.2 ±17.6 | -0.1 ± 0.3 | -0.2 ± 0.4 | -21.0 ±27.5 | 12.9 ±26.1 |
| 0.2 | -13.6 ±14.4 | 0.1 ± 0.3 | -0.2 ± 0.4 | 4.8 ±27.5 | -5.8 ±26.1 |
| 0.3 | -12.8 ±12.0 | -0.2 ± 0.2 | -0.2 ± 0.3 | -1.2 ±27.4 | 0.5 ±26.1 |
| 0.4 | 2.9 ±10.1 | -0.1 ± 0.2 | -0.1 ± 0.3 | -21.1 ±27.4 | 23.1 ±26.0 |
| 0.5 | 0.5 ± 8.6 | -0.1 ± 0.2 | -0.2 ± 0.3 | 17.8 ±27.3 | -8.3 ±26.0 |
| 0.6 | 25.2 ± 7.3 | 0.1 ± 0.2 | -0.3 ± 0.2 | 5.7 ±27.3 | 6.6 ±25.9 |
| 0.7 | 25.6 ± 6.1 | -0.1 ± 0.2 | 0.1 ± 0.2 | 16.8 ±27.3 | -7.2 ±25.8 |
| 0.8 | -0.9 ± 5.0 | -0.2 ± 0.1 | 0.1 ± 0.2 | -5.8 ±27.3 | 23.3 ±25.7 |
| 0.9 | 13.8 ± 3.7 | 0.1 ± 0.1 | 0.2 ± 0.1 | -10.2 ±27.2 | 14.5 ±25.5 |

Table 5: Residuals between total annual posterior fluxes and total annual true fluxes for the five simulated species (in MtC/yr) and different inter-species correlation values in the prior error covariance matrix (first column). The corresponding posterior uncertainty was added for each Post-Truth value.

**-- Note to the Referee: the values for the correlation of 0.7 do not exactly reproduces the values in Table 4, as we realized that in the uploaded version of the paper, the zi-correction described in section 2.1.1 was mistakenly turned off. This has been fixed in the revised version, and the updated Table 4 has values matching the 7th row of Table 5.**

*For all of the experiments, the residuals between true and posterior fluxes are lower than residuals between true and prior fluxes for each of the simulated species; the difference between the cases with maximum and minimum residuals is around 4.2%. In addition, we found that the posterior aggregated fluxes in the nine experiments are not significantly different from each other, implying that the system is fairly robust against errors in the assumed inter-species correlation.*

2. Discussions are more focused on the domain total. It is interesting to see how well the system will reproduce their spatial distribution.
**1) Note that we do actually not focus on the domain total, as we believe it is not reasonable to constrain the whole European domain when pseudo-observations are focused only around a single city; for this reason we chose the region marked by the 50% footprint area, that contains most of the surface influence. We suggest to add the following sentence at page 8 – line 30:**
*As the pseudo-observations are clustered around a single location (Frankfurt), fluxes over the whole European domain can very likely not be constrained. Therefore, as spatial aggregation scale we chose a domain...*
**2) Regarding the reproduction of spatial distribution: Our modeling framework does not optimize the emissions in the individual grid-cells, but only the scaling factors for emissions from different sectors and fuel types. With this modeling framework it is not possible for us to evaluate how well the spatial distribution is reproduced.**

Minor comments:
1. Line 5, Page 4: "This synergy follows from the fact . . ." Better changed to 'follows the fact. . .' or other phrase.
**The text was edited according to the suggestion**

2. Line 5, Page 14: ". . .have a magnitude of 6-11 Megatons of carbon per year (MtC y-1) in July" The unit of MtC/yr seems inconsistent with annual total presented in Table 5. I think it should be MtC/a.

**In Table 4 (not 5), the total presented refers to overall residuals between (e.g.) total prior fluxes minus total true fluxes, aggregated over all emission categories. As such, they are not directly comparable with the amounts shown in Figure 7 (to which Line 5, Page 14 refers), which instead indicates the true fluxes for specific emission categories. Note that we chose to use the unit "MtC y$^{-1}$" over "MtC/a" as suggested by the journal 'Manuscript preparation guidelines for authors'.**

3. Line 26, page 14: ".. .for the whole year between the prior and both posterior and the perturbed prior" The sentence is unclear.

**The text was modified as follows:**

*"To do so, for each of the five simulated species we calculated the total annual fluxes for prior, posterior, truth, and perturbed prior. From these total fluxes we then derive the overall residual between prior and truth, posterior and truth, and perturbed prior and truth."*

4. Figure 7: Please explain why for CH4 fluxes in December, their uncertainty has been significantly reduced, but the differences from the 'true' are not obviously improved.

**It is normal for Bayesian inversion to have some elements of the posterior state space that are not obviously improved. The expectation is that the posterior values are in agreement with the true values within their respective uncertainty. As we use 1-sigma uncertainties, we expect about 36% to be even outside this uncertainty range. Note that the inversion for monthly fluxes solves for a total of 828 scaling factors for CH$_4$, of which about 70 contribute to 90% of the fluxes. However, the atmospheric signals associated with these 70 different sectors/fuel types are not observed directly, but only as a combined signal in CH$_4$.**

5. Figure 6: I suggest the authors also provide the prior and posterior error correlation between CO2 fossil fuel emission and biospheric net flux in the main text.

**A sentence was added to provide the correlations (Line 26, page 26)**

*Note that CO$_2$ from anthropogenic emissions is assumed to be independent from biogenic emissions; therefore prior error correlation between these categories is zero.*

6. Figure 7: check the units for monthly fluxes (in main text as well).

**The units in Figure 7 and 8 were changed to MtC/m. References to these figures in the main text were also given with this unit.**

---

## Author Comment (AC3) · 15 Dec 2017

Dear Referee #3

Thank you for thoroughly reading and commenting the manuscript. Please find below the replies to your suggestions; each of your suggestions is followed by the corresponding reply in bold letters and (where appropriate) the actions taken to address it in the updated version of the manuscript in italics.

General comments:

The paper presents a multi-species inversion framework tested using pseudo-data experiments. Various assumptions are made to evaluate the sensitivity of the inversion, with an emphasis on the impact of error correlations across species and sectors. Overall, the paper presents an innovative approach to assimilate various atmospheric species in a single inversion framework. This study is clearly worthwhile publishing but lacks a better evaluation of the aggregation operator assumption (perfect prior emission distribution) and the impact of systematic errors in the system affecting the correlations in the gas-sector attribution problem. The Observing System Simulation Experiments (OSSE's) cover some of the assumptions with varying levels of uncertainties but several components are not carefully considered. The two major concerns here are the aggregation operator, that remains perfectly known and so the spatial distribution of the prior fluxes, and the assessment of correlations among sectors and across trace gases for the different species that remain very unclear. A last but less critical concern is related to the assumption that transport errors are similar across species, which is unlikely for CH4 and CO2 for example, rarely co-emitted (only CO2-CO is discussed) and therefore affected by different problems in different parts of the domain. The work focuses primarily on random errors and ignores systematic errors that remain the main limitations in atmospheric inversions. Therefore, this study requires some additional experiments before publication, specifically addressing the error associated with the aggregation operator and errors in gas ratios for the different sectors.

- The use of an aggregation operator needs to be discussed. Hyper-parameters (here scaling factors for the sectors) are used to reduce the dimension of the problem but correspond to an assumption of perfectly-known distributions. The system should be evaluated not only under the "perfect spatial distribution" assumption, especially for CO2 biogenic fluxes which are clearly not well-known. One suggestion to clarify the concern here would be to use VPRM as truth but assumes a different distribution when constructing the aggregated solution such as the posterior fluxes from Panagiotis et al. (2016). Other experiments could be designed here to test the aggregation problem. Similarly, the area defined by half of the total footprint is arbitrary and never tested nor justified. Why 50% was used? How much variations are expected within that area which would affect the error correlations? If a power plant is located near an airport, how would that affect the CO/CO2 correlations and therefore the homogeneity within the aggregated area?

**Note that we do actually not focus on the domain total, as we believe it is not reasonable to constrain the whole European domain when pseudo-observations are focused only around a single city; for this reason we chose the region marked by the 50% footprint area, that contains most of the surface influence. We suggest to add the following sentence at page 8 – line 30:**

*… into physically representative quantities. As the pseudo-observations are clustered around a single location (Frankfurt), fluxes over the whole European domain can very likely not be constrained. Therefore, as spatial aggregation scale we chose a domain…*

**To the main point of this comment, we actually do not exactly assume perfect knowledge of the spatial distribution of total emissions; it is only within each sector and fuel type the spatial pattern of the emissions are assumed to be known.**

**We admit that the modeling framework that we set up is not particularly well suited to investigate the aggregation error. However, the chosen domain is quite small, and the total fossil fuel fluxes are divided according to species, emission categories, fuel types and months. This result in numerous degrees of freedom available to resolve biosphere fluxes, and for this reason we expect the aggregation error not to be a particularly important source of uncertainty.**

**In our inversion, as in all inversions, the near field is a critical domain in the arising of systematic errors. The better way to address systematic errors is of course by comparing model outputs with real observations, which are currently unavailable. The bias errors in atmospheric inversions making use of airborne measurements will have to be addressed anyway, once real observations from IAGOS will be available. For this reason, in this paper we chose to focus on random errors instead.**

**We suggest to add the following sentence at page 8 – line 31:**
*… 2011 (Fig. 1). Note that by using this aggregation scale we assume perfectly-known distribution within a given flux category that can result in aggregation error, especially with respect to biogenic fluxes, that are not so well known as anthropogenic fluxes. However, the chosen domain of aggregation is quite small, and the total anthropogenic fluxes are divided according to species, emission categories, fuel types and months. This result in 69 numerous degrees of freedom per month for each anthropogenic species and 10 degrees of freedom per month for the biospheric fluxes; for this reason we expect the aggregation error not to be a particularly important source of uncertainty.*

- The discussion about error correlations across species is confusing. How did you define the emissions for the different sectors? Have you assigned gas ratios to various sectors? If so, what are these ratios? Some of the discussions are

related to using CO2 and CO data to diagnose gas-to-gas correlations, but the exact definition of the emissions of the different gases for each sector has been defined in the inversion system. Or maybe the sectors are unrelated for each gas? The different sectors have ratios in terms of trace gas emissions but these emission ratios vary regionally. This section needs to be explained in more details. The assumptions made here should also be tested in the inversion framework.

**Emission ratios are not used here, but we used instead bottom-up calculated emissions for each of the three gases, using different emission sector-specific factors, which are for CO also region-specific. These country emissions are then gridded consistently with geospatial proxy data that are representative for the emitting activity, common to all species for the multi-species sources.**

**We suggest the following changes to the text:**

**Add at page 7, line 1:**
*…on our regional European domain. For each of the three anthropogenic modeled species ($CO_2$, CO and $CH_4$), different emission maps are used as input. Temporal profiles are then applied to these sector- and fuel-specific emission maps.*

**Replace at page 14, from line 5 to line 24:**
*CO2 and CO are dominated by combustion sectors. The most important emission sectors for CO2 are energy, industry, transport and building, each contributing 7-10 MtC y-1 in July and 6-14 MtC y-1 in December. Dominant fuels for CO2 are coal, gas and oil, whose prior fluxes (pseudo data) have a magnitude of 6-11 Megatons of carbon per year (MtC y-1) in July and 8-14 MtC y-1 in December. For CO the most important emission sector is heating of buildings during winter contributing a 0.19 MtC y-1 flux with only secondary contributions from industry and transport with a magnitude of 0.04 MtC y-1 and 0.05 MtC y-1 respectively (during July and December). The dominant fuel for CO is biofuel with 0.19 MtC y-1 emissions during winter. The secondary industrial and transport contributions originate in summer from oil and biofuels with a magnitude of 0.06-0.08 MtC y-1 and from agricultural waste burning with a magnitude of 0.06-0.11 MtC y-1.*

*Contrary to CO2 and CO, CH4 is determined by non-combustion sectors, more specifically by a contribution of 0.15 MtC y-1 flux from agriculture (manure management and rice cultivation) in July with secondary contributions from waste and energy with a magnitude of roughly 0.06-0.08 MtC y-1 in both July and December. Other non-combustion sectors, in particular wastewater treatment and landfills contribute to a total of 0.16-0.24 MtC y-1 of emissions. These non-combustion sectors contribute to less than 20% of total CO2 emissions, with 1.13 MtC y-1 from the cement and lime industry and less than 20% to the total CO emissions (0.03 MtC y-1 from the metal industry).*
*The contribution to $CO_2$ from biospheric primary production (a sink for atmospheric $CO_2$) is about 100 MtC y-1 in July, which drops to almost zero in*

*December, while respiration values are 50 MtC y-1 in July and roughly 150 MtC y-1 in December.*

- CO biogenic fluxes: the paper does not address the problem of CO biogenic fluxes during the growing season. Warm days in summer correspond to large amount of biogenic VOC's being emitted from the vegetation, producing CO to non-negligible levels. This issue should be discussed if not addressed. How would this problem affect the ability to retrieve the truth?

**To discuss this issue we propose to add the following at page 14, line 4:**

*Note that our modeling framework does not allow for simulating CO biogenic fluxes during the growing season. Warm days in summer correspond to large amount of biogenic VOC's being emitted from the vegetation, producing CO to non-negligible levels. According to Hudman (2008), anthropogenic emissions accounts for only 31% of CO emissions in the US during summer. Conversely, according to estimates from EDGAR, CO anthropogenic emissions during summer are about 18% of the annual anthropogenic emissions. Combining these two results, one could conclude that CO production from biogenic sources accounts for roughly 42% of total annual CO emissions.*

*In general, the absence of some emission sources in an inventory is equivalent to the assumption of having point sources not included in the emission map, but still contributing to the measurements. The inversion scheme would typically react to this by assigning such point sources in some other sector other fuel type. As a result, the posterior enhancements would be biased low in proximity of that point sources, and (slightly) biased high for influences from other regions with the same sector or fuel type. This issue should definitely be considered in a future study making use of actual CO, $CO_2$ and $CH_4$ observations from IAGOS but has limited effects on this paper, as our main focus is on the benefits of inter-species correlation on the posterior uncertainty in the frame of a synthetic experiment.*

- When you constructed your error correlations for CH4, transport errors are unlikely to be highly correlated as CH4 is only partially co-emitted with CO2 and CO. Large emissions from NG production and farming activities are uncorrelated with biogenic or fossil fuel consumption. This problem should be addressed here. If transport errors, which are spatially variable, affect CH4 and CO2/CO in different ways, the error correlation would be affected. Additional experiments using incorrect error correlations would quantify the sensitivity of the inverse fluxes to the assumptions made in prior errors.

**This is a very useful suggestion, which we followed now. We propose to add the following at Page 14, Line 29**

*"Improper characterization of the error correlation may result in systematic bias in the posterior estimate. As mentioned in Sect. 2.1.6, inter-species correlation, the correlation between different fuel types and the correlation between different emission sectors in **Sprior** is assumed equal to 0.7 (Sect. 2.1.4). To assess how well the system will reproduce the 'true' fluxes with incorrectly specified correlations, a series of experiment was performed in which the inter-species correlation in*

*Sepsilon* remains equal to 0.7, while the three correlation coefficients in **Sprior** assume different values ranging from 0.1 to 0.9. Table 5 shows the residuals between total annual posterior fluxes and total annual true fluxes for the five simulated species, derived similarly as for Table 4.
*We found that for all species the uncertainty reduction increases with correlation. More precisely, from correlation 0.1 to 0.9, the annual uncertainty reduction for anthropogenic $CO_2$ increases from 6.5% to 40.9%, while the increase is lower for GEE (from 64.6% to 65.2%) and respiration (from 35.1% to 36.8%) because the biospheric fluxes are independent from other species. For CO, the uncertainty reduction increases from 40.6% (with correlation 0.1) to 57.5 (with correlation 0.9). The annual uncertainty reduction for CH4 increases from 32.6% to 59.0%.*

*In addition, the posterior-truth biases are always lower than the prior-truth biases. The posterior uncertainty values are almost always larger then the corresponding bias values, except for CO with prior correlation equal 0.8, and fossil fuels $CO_2$ with prior correlations equal to 0.6, 0.7 and 0.9. Thus, except for these few cases, the posterior is not significantly different from the truth. Conversely, the prior (not shown) is significantly different than the truth in the majority of cases for fossil fuel fluxes, and in some cases also for biogenic fluxes. The effect of assuming the incorrect error correlations appears to be in general small. Following this result, the fact that $CH_4$ is only partially co-emitted with $CO_2$ and CO should not affect the inversion in a strong way.*

| Correlation | Post-Truth $CO_2$ ff | Post-Truth CO | Post-Truth $CH_4$ | Post-Truth GEE | Post-Truth Respiration |
|---|---|---|---|---|---|
| 0.1 | -17.2 ±17.6 | -0.1 ± 0.3 | -0.2 ± 0.4 | -21.0 ±27.5 | 12.9 ±26.1 |
| 0.2 | -13.6 ±14.4 | 0.1 ± 0.3 | -0.2 ± 0.4 | 4.8 ±27.5 | -5.8 ±26.1 |
| 0.3 | -12.8 ±12.0 | -0.2 ± 0.2 | -0.2 ± 0.3 | -1.2 ±27.4 | 0.5 ±26.1 |
| 0.4 | 2.9 ±10.1 | -0.1 ± 0.2 | -0.1 ± 0.3 | -21.1 ±27.4 | 23.1 ±26.0 |
| 0.5 | 0.5 ± 8.6 | -0.1 ± 0.2 | -0.2 ± 0.3 | 17.8 ±27.3 | -8.3 ±26.0 |
| 0.6 | 25.2 ± 7.3 | 0.1 ± 0.2 | -0.3 ± 0.2 | 5.7 ±27.3 | 6.6 ±25.9 |
| 0.7 | 25.6 ± 6.1 | -0.1 ± 0.2 | 0.1 ± 0.2 | 16.8 ±27.3 | -7.2 ±25.8 |
| 0.8 | -0.9 ± 5.0 | -0.2 ± 0.1 | 0.1 ± 0.2 | -5.8 ±27.3 | 23.3 ±25.7 |
| 0.9 | 13.8 ± 3.7 | 0.1 ± 0.1 | 0.2 ± 0.1 | -10.2 ±27.2 | 14.5 ±25.5 |

Table 5: Residuals between total annual posterior fluxes and total annual true fluxes for the five simulated species (in MtC/yr) and different inter-species correlation values in the prior error covariance matrix (first column). The corresponding posterior uncertainty was added for each Post-Truth value.

- The problem of unreported sources in CH4 inventory is not addressed at all. Recent papers have discussed the lack of information for natural gas and oil production operations, or from recent and old mining areas. How would

unreported sources affect the inverse solutions? This question comes back to the aggregation operator.

**To discuss this issue we propose to add the following at page 14, line 4:**

*Our modeling framework is currently not well suited to account for unreported sources of $CH_4$ due to the lack of information about natural gas and oil production operations, or from recent and old mining areas.. Many recent studies have discussed the problem, mainly referring to shale basins exploited via hydraulic fracturing in the US (e.g. Kort et al., 2016; Karion et. al, 2015; Lyon et al., 2015). For example, Karion (2015) concludes that EDGAR underestimates methane emissions associated with oil and gas industry by a factor of 5 in the US.*

*However, the situation over the European continent may be quite different. In a review about risk assessment of shale gas development in the UK, Prpich (2015) reports that the European Union is generally much more cautious about unconventional oil and gas sources, while a recent study on a methane plume over the North Sea (Cain et al., 2017) concluded that the bulk signature of said plume originated from on-shore coal mines and power stations in the Yorkshire area.*

*In general, the absence of some emission sources in an inventory is equivalent to the assumption of having point sources not included in the emission map, but still contributing to the measurements. The inversion scheme would typically react to this by assigning such point sources in some other sector other fuel type. As a result, the posterior enhancements would be biased low in proximity of that point sources, and (slightly) biased high for influences from other regions with the same sector or fuel type. This issue should definitely be considered in a future study making use of actual CO, $CO_2$ and $CH_4$ observations from IAGOS but has limited effects on this paper, as our main focus is on the benefits of inter-species correlation on the posterior uncertainty in the frame of a synthetic experiment*

- The utility of the figures showing the multiple error covariance matrices for the different cases remains limited. The information content would be better described with words or mathematically. Readers cannot extract useful information from contour plots of covariance matrices. They could remain part of the paper but as part of the supplementary information. A table could also synthesize the various assumptions tested in the inversion system.

**We propose to add axis label to Fig. 2,3,4 to increase readability. Such axis should identify different species, emission sectors, fuel types and vegetation categories. In addition, we suggest to introduce two different equations (see below) to describe mathematically the error structure in the different cases.**

*Eq6.2: $S_{multi}$ =*

$$\begin{matrix} X_{co,co} & X_{co,co2} & X_{co,ch4} \\ X_{co2,co} & X_{co2,co2} & X_{co2,ch4} \\ X_{ch4,co} & X_{ch4,co2} & X_{ch4,ch4} \end{matrix}$$

*Eq6.3: $S_{single}$ =*

$$\begin{matrix} X_{co,co} & 0 & 0 \\ 0 & X_{co2,co2} & 0 \\ 0 & 0 & X_{ch4,ch4} \end{matrix}$$

*Note that each element in Eq 6.2 and 6.3 is a sub-matrix. In the case of $S_{prior}$, each element of such sub-matrices indicates the covariance between different flux categories. Conversely, in the case of $S_{epsilon}$ , the sub-matrices show the covariance between different observations.*

Technical comments:

3-1: Consequently, intercomparisons...
**The text was edited according to the suggestion**

3-3: the international level
**The text was edited according to the suggestion**

3- 1st paragraph: This paragraph is confusing and not always following a logical path. Prediction skills and emission reduction are two different problems not directly connected to each other. Explain better the broad context of this study by focusing on the main general issues and clarify which one you are trying to address here.
**The paragraph was rephrased as follows:**
*As widely recognized at the international level, there is a need for reduction in anthropogenic emissions (IPCC). This however implies the necessity for reliable climate predictions from atmospheric models in order to allow policymakers to take informed decisions. Unfortunately, current climate predictions are hampered by excessive uncertainties; for example intercomparisons of different models show important differences on their predictions as shown in Friedlingstein (2016). This makes it difficult to assess the better environmental policies to implement. Because most biogenic fluxes ...*

3-10: A commonly used approach to estimate...
**The text was edited according to the suggestion**

3-13: Actually, the uncertainty reduction relies purely on the assumptions made in the system and not on the effective ability of the system to produce a reliable solution. Bayesian system assumes that data will improve the a priori by construction. Explain better what you mean here.
**The text was modified as follows.**
*As the main goal of this study is to assess the benefit of inter-species correlations in reducing the uncertainty of the posterior state space, we are particularly interested*

*in the effects of such correlations on the uncertainty reduction, defined as the difference between prior and posterior uncertainty normalized by the prior.*

3- 2nd paragraph: Several papers are missing here. For example, CO2-CH4 inversion using satellite data (Pandey et al., 2015) or the optimization of co-emitted species (Brioude et al., 2012), and early work on delta 13-CO2 by Enting et al. (1995). The authors should dig into atmospheric chemistry studies where several studies have addressed the use of multiple co-emitted species to constrain emissions at small scales.

Previous studies using multiple species to constrain emissions should be introduced here, even without having used a formal inversion framework, such as urban studies over Los Angeles (e.g. Peischl et al., 2013). The optimization problem is equivalent and relies on similar ideas to constrain the emissions.

**We replaced the text at Pag. 3, lines 20-22 with:**

*Several studies have made use the correlations among different species. One of the first example is the work from Enting (1995) on $CO_2$ and $^{13}CO_2$, while Brioude (2012) attempted to derive a $CO_2$ emission inventory without a prior emission estimate, instead using inventories of CO, $NO_y$ and $SO_2$. Similarly, Peischl (2013) made use of CO and $CO_2$ inventories to help quantifying sources of $CH_4$ in the Los Angeles basin. The ability of measuring multiple species has been proved useful also in remote sensing. For example, Pandey (2015) made use of simultaneously retrieved $CO_2$ and $CH_4$ total column to reduce scattering effect. Further examples of studies making use of co-emitted species can be found in the frame of atmospheric chemistry (Konovalov et al., 2014; Berezin et al., 2013; Pison et al., 2009). More focused on exploiting inter-species correlation to reduce uncertainty in the frame of Bayesian Inversion, Palmer (2006) made use of CO2-CO correlations to improve an inversion using data from the TRACE-P aircraft mission, while Wang (2009) employed a similar method using satellite data, obtaining a reduction in the flux error of a $CO_2$ inversion.*

5-24: This technique assumes that the wind direction and speed are comparable near the surface and at 2km high. Mass-balance studies have shown that this is often not the case (e.g. Karion et al., 2015). Free troposheric air represents different air masses due to the wind direction and speed gradients in the vertical. This assumption would need to be tested with the particle model.

**This is a misunderstanding. We do not rely on winds within the mixed layer and the wind above to be comparable, as our transport operator H represents the mixed layer enhancements appropriately. We added the following text to**

*... a single footprint is derived. To represent the mixed layer enhancements, the footprints for receptors within the boundary layer are averaged, and the footprint for the free tropospheric receptor is subtrated from this, resulting in a footprint for the mixed layer enhancements. This footprint is then matrix-multiplied ...*

7-3: What about CO biogenic fluxes? During warm summer times, biogenic CO

fluxes represent a significant fraction of the signals. Did you ignore this contribution in your study?

**A similar comment has already been addressed (see above).**

2.1.3 To reduce the dimension of the state vector, you assume here that the spatial distribution of the prior fluxes and emissions are perfect, using an aggregation operator. This approach is reasonable for fossil fuel emissions but less convincing for biogenic fluxes.

**A similar comment has already been addressed (see above).**

12-28: How did you take into account the truncation of the prior errors? Did you adjust the truncated random perturbations to match the non-truncated assumption made in the prior error covariance matrix?

**The error realization is obtained by multiplying a randomly generated, normally distributed vector with the prior error covariance matrix. This ensures that such realization has the same error correlation of the prior uncertainty. Where the result of such matrix-vector product is negative, the same operation is performed recursively until all elements of the state vector are positive. We suggest adding the following text at page 12 line 24:**

*… to avoid negative state vector values. In detail, the error realization is obtained by multiplying a randomly generated, normally distributed vector with the prior error covariance matrix. This ensures that such realization has the same error correlation of the prior uncertainty. Where the result of such matrix-vector product is negative, the same operation is performed recursively until all elements of the state vector are positive. This ensures that the difference …*